# Colonoscopy and fecal immunochemical testing versus usual care in diagnostic colorectal cancer screening: the SCREESCO randomized controlled trial

There is a need to quantify the benefits and harms of colorectal cancer (CRC) screening using primary colonoscopy or fecal immunochemical testing (FIT) compared with usual care with no screening. Guidelines recommend screening in individuals aged 50–75 years using colonoscopy or FIT, and many screening programs use one-sample biennial FIT. Here we compare incidence of diagnosed CRCs and gastrointestinal and cardiovascular events between screening and usual care during the diagnostic phase of the SCREESCO trial. A randomized block method (no masking) assigned 278,280 individuals aged 60 years to once-only colonoscopy, 2 rounds of two-stool FIT with a low cutoff (10 μg g$^{-1}$ feces) or usual care (control group) in a ratio of 1:6 for colonoscopy versus control and 1:2 for FIT versus control. In the analysis, 31,113 individuals were in the primary colonoscopy arm and 60,267 were in the FIT arm, and there were 186,671 primary colonoscopy controls, of whom 120,521 were also controls for comparison with the FIT arm. After a median follow-up of 4.8 years, the incidence rate of CRC was 107.9 in the colonoscopy arm and 99.9 in controls per 100,000 person-years (incidence rate ratio (IRR): 1.08, 95% confidence interval (CI): 0.91–1.28) and 96.0 in the FIT arm and 103.9 in controls (IRR: 0.92, 95% CI: 0.81–1.05). Rates of stage I–II CRC were higher in the colonoscopy arm (IRR: 1.38, 95% CI: 1.09–1.74) and in the FIT arm (IRR: 1.19, 95% CI: 0.99–1.43) versus controls. Rates of cardiovascular and gastrointestinal events were slightly higher in the intervention arms during the first year and were subsequently more similar to controls. Our findings of an increase in CRC detection implies a benefit of screening while the increase in adverse events suggests some initial harm. ClinicalTrials.gov: NCT02078804.

To inform health policy in early detection and removal of CRC and precancerous lesions, there is a need to quantify the benefits and harms from CRC screening. The American College of Gastroenterology and the European Society of Gastrointestinal Endoscopy recommend CRC screening in individuals aged 50–75 years by colonoscopy or FIT, where colonoscopy is performed only in those with increased risk indicated by a higher fecal hemoglobin level[1,2]. Several countries have implemented FIT-based organized screening often with a similar age range and biennial testing but with cutoffs for a positive test ranging between 8.5 μg hemoglobin per g feces and 120 μg hemoglobin per g feces[3,4]. In Sweden, screening by use of a one-sample biennial FIT has been ongoing in the Stockholm-Gotland region since 2015 in individuals aged 60–69 years and since 2020 in individuals aged 60–74 years, with a cutoff of 40 μg hemoglobin per g feces for women and 80 μg hemoglobin per g feces for men[5,6]. A national biennial FIT-based screening program with the same cutoffs and age range began its rollout in 2021 with full implementation planned for 2026.

✉e-mail: marcus.westerberg@uu.se

**Fig. 1 | Study flowchart.** Number of randomized, excluded and included individuals.

There are several trials of colonoscopy screening, but few have examined FIT compared to usual care[7–14]. The Spanish randomized controlled trial (RCT) COLONPREV recently found similar CRC incidence in the FIT arm and the primary colonoscopy arm at 10 years but did not include any control arm[15]. Colonoscopy may cause serious adverse events, although the reported rates have been low[8,16]. There is also limited evidence from RCTs on the impact of screening using primary colonoscopy and FIT on CRC incidence and adverse events compared to usual care.

The RCT SCREESCO trial (Screening of Swedish Colons; ClinicalTrials.gov: NCT02078804) in Sweden includes a primary colonoscopy screening arm, two rounds of a two-stool FIT screening arm with an unusually low cutoff (10 µg g⁻¹ feces) and a control arm with individuals not invited to screening but following usual care[17]. By adding extensive information from Swedish healthcare registers, we now report a comparison of the diagnostic yield, in terms of total CRCs diagnosed, and of adverse events occurring in all participants in the trial—screening versus control arm—during the diagnostic phase (from 2014 to 2020, when all interventions occurred) on an intention-to-screen basis. The aim of the present study was to assess if the randomization has resulted in the three arms being similar in terms of baseline demographic variables and medical history and to assess if the screening approaches resulted in a higher rate of diagnosis of CRC during the diagnostic phase, especially stage I–II CRC, compared to usual care. It also aims to assess potential harms of screening approaches in terms of an increased rate of cardiovascular and gastrointestinal events in general and short-term all-cause mortality during the diagnostic phase. The present analysis extends previous trial information, which was limited to screen-detected CRCs and adverse events in screening colonoscopies[17].

## Results

### Patient disposition

Between 11 February 2014 and 1 March 2016, 201,000 individuals turning 60 or aged 60 were randomized to one of three arms: primary colonoscopy, two rounds of two-stool FIT (FIT×2) or controls. All these controls were controls to both intervention arms. Because of low participation in the primary colonoscopy arm, an additional 77,280 individuals were randomized to primary colonoscopy or control between 30 May 2017 and 25 May 2018. A total of 278,280 individuals were randomized. Due to administrative lag in registration, 159 individuals who died and 70

individuals with prevalent CRC before the date of randomization were identified only after randomization and were excluded. Another two control individuals were excluded because they were not identifiable in any register (Fig. 1). There were 278,051 unique individuals included in the final analyses, 31,113 in the primary colonoscopy arm, 60,267 in the FIT×2 arm and 186,671 primary colonoscopy controls (randomized 2014–2018), of whom 120,521 were also controls for comparison with the FIT×2 arm (that is, FIT×2 controls, randomized in 2014–2016).

Baseline demographic variables (for example, educational level and country of birth) and history of comorbidities (Table 1) were balanced between the primary colonoscopy arm and colonoscopy controls (n = 186,671) and between the FIT×2 arm and FIT×2 controls. Most individuals (92%) did not have any prior gastrointestinal or cardiovascular event.

The maximum and median follow-up were 6.9 years and 4.8 years, respectively, and 1,385 (0.5%) individuals migrated out of Sweden during the follow-up. The median time from randomization to screening colonoscopy in the 10,679 (35% of the 30,400 invited) participants in the primary colonoscopy arm was 349 days (interquartile range (IQR): 253–446 days), and the median time from randomization to first FIT in the 33,383 (55% of the 60,137 invited) participants in the FIT×2 arm was 338 days (IQR: 229–395 days) (Extended Data Fig. 1). Men (35%) participated slightly more often than women (33%) in the primary colonoscopy arm, whereas more women (59%) participated in the FIT×2 arm than men (52%) (Supplementary Table 1). Individuals who did not participate in colonoscopy in the primary colonoscopy arm or in fecal testing in the FIT×2 arm (that is, non-participants) had lower educational level, originated more often from another country and exhibited slightly more comorbidity.

### Diagnostic yield of screening and usual care

The incidence rate of diagnosed CRCs during the diagnostic phase was 107.9 per 100,000 person-years (160 individuals/148,337 person-years) in the primary colonoscopy arm and 99.9 (889 individuals/889,774 person-years) in the control arm (incidence rate ratio (IRR): 1.08, 95% confidence interval (CI): 0.91–1.28) and 96.0 (330 individuals/343,703 person-years) in the FIT×2 arm and 103.9 (715 individuals/687,048 person-years) in the FIT×2 control arm (IRR: 0.92, 95% CI: 0.81–1.05) (Table 2). Figure 2 shows that the incidence rates and IRRs peaked

**Table 1 | Baseline characteristics of all individuals included (intention to screen)**

| | Primary colonoscopy arm | | Control arm | | FIT×2 arm | | FIT×2 control arm[a] | |
|---|---|---|---|---|---|---|---|---|
| | N | (%) | N | (%) | N | (%) | N | (%) |
| N individuals | 31,113 | (100.0) | 186,671 | (100.0) | 60,267 | (100.0) | 120,521 | (100.0) |
| Sex | | | | | | | | |
| Men | 15,550 | (50.0) | 93,309 | (50.0) | 30,128 | (50.0) | 60,261 | (50.0) |
| Women | 15,563 | (50.0) | 93,362 | (50.0) | 30,139 | (50.0) | 60,260 | (50.0) |
| Year of randomization | | | | | | | | |
| 2014 | 6,697 | (21.5) | 40,171 | (21.5) | 20,091 | (33.3) | 40,171 | (33.3) |
| 2015 | 6,696 | (21.5) | 40,173 | (21.5) | 20,088 | (33.3) | 40,173 | (33.3) |
| 2016 | 6,693 | (21.5) | 40,177 | (21.5) | 20,088 | (33.3) | 40,177 | (33.3) |
| 2017 | 5,511 | (17.7) | 33,061 | (17.7) | 0 | (0.0) | 0 | (0.0) |
| 2018 | 5,516 | (17.7) | 33,089 | (17.7) | 0 | (0.0) | 0 | (0.0) |
| Healthcare region of residence | | | | | | | | |
| North | 3,013 | (9.7) | 18,074 | (9.7) | 5783 | (9.6) | 11,563 | (9.6) |
| Central | 8,806 | (28.3) | 52,834 | (28.3) | 17,444 | (28.9) | 34,883 | (28.9) |
| Southeast | 4,420 | (14.2) | 26,514 | (14.2) | 8484 | (14.1) | 16,960 | (14.1) |
| South | 6,774 | (21.8) | 40,613 | (21.8) | 12,993 | (21.6) | 25,988 | (21.6) |
| West | 8,100 | (26.0) | 48,636 | (26.1) | 15,563 | (25.8) | 31,127 | (25.8) |
| Educational level | | | | | | | | |
| Low | 5,704 | (18.3) | 34,378 | (18.4) | 11,569 | (19.2) | 23,134 | (19.2) |
| Intermediate | 15,227 | (48.9) | 92,111 | (49.3) | 29,254 | (48.5) | 58,774 | (48.8) |
| High | 10,093 | (32.4) | 59,669 | (32.0) | 19,253 | (31.9) | 38,262 | (31.7) |
| Missing | 89 | (0.29) | 513 | (0.27) | 191 | (0.32) | 351 | (0.29) |
| Country of birth | | | | | | | | |
| Sweden | 27,376 | (88.0) | 163,930 | (87.8) | 53,154 | (88.2) | 106,235 | (88.1) |
| Other | 3,737 | (12.0) | 22,741 | (12.2) | 7,113 | (11.8) | 14,286 | (11.9) |
| Charlson Comorbidity Index | | | | | | | | |
| 0 | 24,838 | (79.8) | 148,532 | (79.6) | 48,129 | (79.9) | 95,995 | (79.7) |
| 1 | 2,951 | (9.5) | 18,299 | (9.8) | 5,849 | (9.7) | 11,812 | (9.8) |
| 2 | 2,316 | (7.4) | 13,867 | (7.4) | 4,380 | (7.3) | 8,922 | (7.4) |
| ≥3 | 1,008 | (3.2) | 5,973 | (3.2) | 1,909 | (3.2) | 3,792 | (3.1) |
| Drug comorbidity index[b] | | | | | | | | |
| <0 | 3,591 | (11.5) | 21,643 | (11.6) | 7,280 | (12.1) | 14,242 | (11.8) |
| 0–0.22 | 11,769 | (37.8) | 71,469 | (38.3) | 22,855 | (37.9) | 46,142 | (38.3) |
| 0.23–0.91 | 7,867 | (25.3) | 46,692 | (25.0) | 15,369 | (25.5) | 30,295 | (25.1) |
| ≥0.92 | 7,886 | (25.3) | 46,867 | (25.1) | 14,763 | (24.5) | 29,842 | (24.8) |
| Any prior cardiovascular or gastrointestinal event | | | | | | | | |
| No | 28,706 | (92.3) | 17,2738 | (92.5) | 55,724 | (92.5) | 111,527 | (92.5) |
| Yes | 2,407 | (7.7) | 13,933 | (7.5) | 4,543 | (7.5) | 8,994 | (7.5) |
| Cardiovascular event | 1,935 | (6.2) | 11,255 | (6.0) | 3,681 | (6.1) | 7,281 | (6.0) |
| Gastrointestinal event | 565 | (1.8) | 3,248 | (1.7) | 1,052 | (1.7) | 2,066 | (1.7) |

[a]Note that all individuals in the FIT×2 control arm are also in the control arm for primary colonoscopy. [b]Cutoffs based on quartiles of the drug comorbidity index.

around year one after randomization in both the primary colonoscopy arm and the FIT×2 arm as compared to respective control arms. The corresponding cumulative incidence proportions at the end of follow-up of CRC were 0.69% (95% CI: 0.66–0.71%) in the primary colonoscopy arm, 0.72% (95% CI: 0.71–0.73%) in the control arm, 0.61% (95% CI: 0.60–0.63%) in the FIT×2 arm and 0.73% (95% CI: 0.72–0.75%) in the FIT×2 control arm (Extended Data Fig. 2).

The incidence rate of diagnosed stage I–II CRCs in the primary colonoscopy arm was 58.7 per 100,000 person-years compared to 42.5 in the control arm (IRR: 1.38, 95% CI: 1.09–1.74) (Table 2 and Fig. 2). Similarly, the rate in the FIT×2 arm was 52.7 and 44.4 in FIT×2 controls (IRR: 1.19, 95% CI: 0.99–1.43). Cumulative incidence proportions were higher in the intervention arms compared to controls throughout most of the study period (Extended Data Fig. 2).

**Table 2 | Incidence rate (per 100,000 person-years) of individuals diagnosed with CRC during the diagnostic phase**

| Stage | Arm | Screen detected | Events | % | Person-years | Rate | 95% CI | IRR[a] | 95% CI |
|---|---|---|---|---|---|---|---|---|---|
| Any colorectal cancer[b] | Primary colonoscopy | Any | 160 | 0.51 | 148,337 | 107.9 | (92.4–125.9) | 1.08 | (0.91–1.28) |
| | | Yes | 51 | 0.16 | | | | | |
| | | No | 109 | 0.35 | | | | | |
| | Control | No | 889 | 0.48 | 889,774 | 99.9 | (93.6–106.7) | 1.00 | |
| | FIT×2 | Any | 330 | 0.55 | 343,703 | 96.0 | (86.2–106.7) | 0.92 | (0.81–1.05) |
| | | Yes | 124 | 0.21 | | | | | |
| | | No | 206 | 0.34 | | | | | |
| | FIT×2 control | No | 714 | 0.59 | 687,048 | 103.9 | (96.6–111.8) | | |
| Stage I–II[c] | Primary colonoscopy | Any | 87 | 0.28 | 148,337 | 58.7 | (47.5–72.4) | 1.38 | (1.09–1.74) |
| | | Yes | 35 | 0.11 | | | | | |
| | | No | 52 | 0.17 | | | | | |
| | Control | No | 378 | 0.20 | 889,774 | 42.5 | (38.4–47.0) | 1.00 | |
| | FIT×2 | Any | 181 | 0.30 | 343,703 | 52.7 | (45.5–60.9) | 1.19 | (0.99–1.43) |
| | | Yes | 89 | 0.15 | | | | | |
| | | No | 92 | 0.15 | | | | | |
| | FIT×2 control | No | 305 | 0.25 | 687,048 | 44.4 | (39.7–49.7) | 1.00 | |
| Stage III–IV | Primary colonoscopy | Any | 70 | 0.22 | 148,337 | 47.2 | (37.3–59.6) | 0.86 | (0.67–1.11) |
| | | Yes | 14 | 0.04 | | | | | |
| | | No | 56 | 0.18 | | | | | |
| | Control | No | 488 | 0.26 | 889,774 | 54.8 | (50.12–59.9) | 1.00 | |
| | FIT×2 | Any | 139 | 0.23 | 343,703 | 40.4 | (34.2–47.8) | 0.71 | (0.58–0.86) |
| | | Yes | 30 | 0.05 | | | | | |
| | | No | 109 | 0.18 | | | | | |
| | FIT×2 control | No | 393 | 0.33 | 687,048 | 57.2 | (51.8–63.1) | 1.00 | |

[a]IRR (rate in intervention arm / rate in control arm). [b]Stage was unknown in 3 (0.01%) individuals in the primary colonoscopy arm, in 23 (0.01%) individuals in the control arm, in 10 (0.02%) individuals in the FIT×2 arm and in 16 (0.01%) individuals in the FIT×2 control arm. [c]Fifteen individuals with N0M0 had unknown T stage, 5 individuals with T1-4N0 had unknown M stage and 19 individuals with T1M0 had unknown N stage.

The rate of diagnosed stage III–IV CRCs was somewhat lower in the primary colonoscopy arm than in the control arm (IRR: 0.86, 95% CI: 0.67–1.11) and lower in the FIT×2 arm than in the FIT×2 control arm (IRR: 0.71, 95% CI: 0.58–0.86) (Table 2 and Fig. 2). Cumulative incidence proportions in the intervention arms were initially similar to those in controls but became lower than controls after around 4 years after randomization (Extended Data Fig. 2).

Of CRCs diagnosed in the primary colonoscopy arm and in the FIT×2 arm, 32% and 38%, respectively, were screen detected (Table 2). Few CRCs were diagnosed outside of the trial in individuals who had undergone a screening colonoscopy in the primary colonoscopy arm and in the FIT×2 arm (nine (0.03%) and eight (0.01%) individuals, respectively) during the diagnostic phase. Two of the 170 previously reported CRCs detected within the trial were, after a review of medical charts, deemed not to be CRC. Nine additional CRCs were similarly detected within the trial after the review. All but two of the 177 CRCs diagnosed within the trial were also registered in the Swedish Colorectal Cancer Register or the Swedish Cancer Register. Stage disagreed in three (1.7%) and was missing in nine (5.1%) of these CRCs (Supplementary Table 2).

Compared to women, men had a higher rate of CRC in the FIT×2 arm (IRR men / IRR women: 1.29, 95% CI: 0.99–1.67) and especially of stage III–IV CRC in both intervention arms (Supplementary Table 3).

### Safety
**Medium-term adverse events.** Incidence rates of gastrointestinal or cardiovascular events were slightly higher in the intervention arms compared to the control arms during the first year of follow-up but

were subsequently more similar (Fig. 3). At the end of follow-up, the incidence rate of cardiovascular events was similar in all arms—that is, 1,475.8 per 100,000 person-years in the primary colonoscopy arm and 1,475.8 per 100,000 person-years in the control arm (IRR: 1.00, 95% CI: 0.96–1.05), although the rate of venous thromboembolism was 60.1 in the FIT×2 arm compared to 43.3 in corresponding controls (IRR: 1.39, 95% CI: 1.16–1.66) (Table 3 and Supplementary Table 4). The rate of gastrointestinal events was somewhat higher in the FIT×2 arm compared to the control arm also at the end of follow-up, primarily iatrogenic bleeding (IRR: 1.18, 95% CI: 1.05–1.32) and unspecified gastrointestinal bleeding (IRR: 1.14, 95% CI: 1.04–1.26).

We previously showed that the proportion of serious adverse events directly linked to screening colonoscopies during the trial was 0.2%, including two bowel perforations and 15 major bleedings[17].

**Death from any cause.** Death (from any cause) was similar in all arms, with a rate of 554.6 per 100,000 person-years (825 deaths) in the primary colonoscopy arm and 579.0 per 100,000 person-years (5,163 deaths) in the control arm (IRR: 0.96, 95% CI: 0.89–1.03) and 577.1 (1,989 deaths) in the FIT×2 arm and 601.8 (4,145 deaths) in the FIT×2 control arm (IRR: 0.96, 95% CI: 0.91–1.01) (Table 3 and Fig. 3). We previously reported one death in the FIT group on day 15 after the colonoscopy, which followed a pulmonary embolism in a man with metastatic cancer[17].

**Adverse events and deaths in men and women.** Men had a similar rate of cardiovascular events, a somewhat lower rate of gastrointestinal events and a higher rate of death than women (Supplementary Table 5).

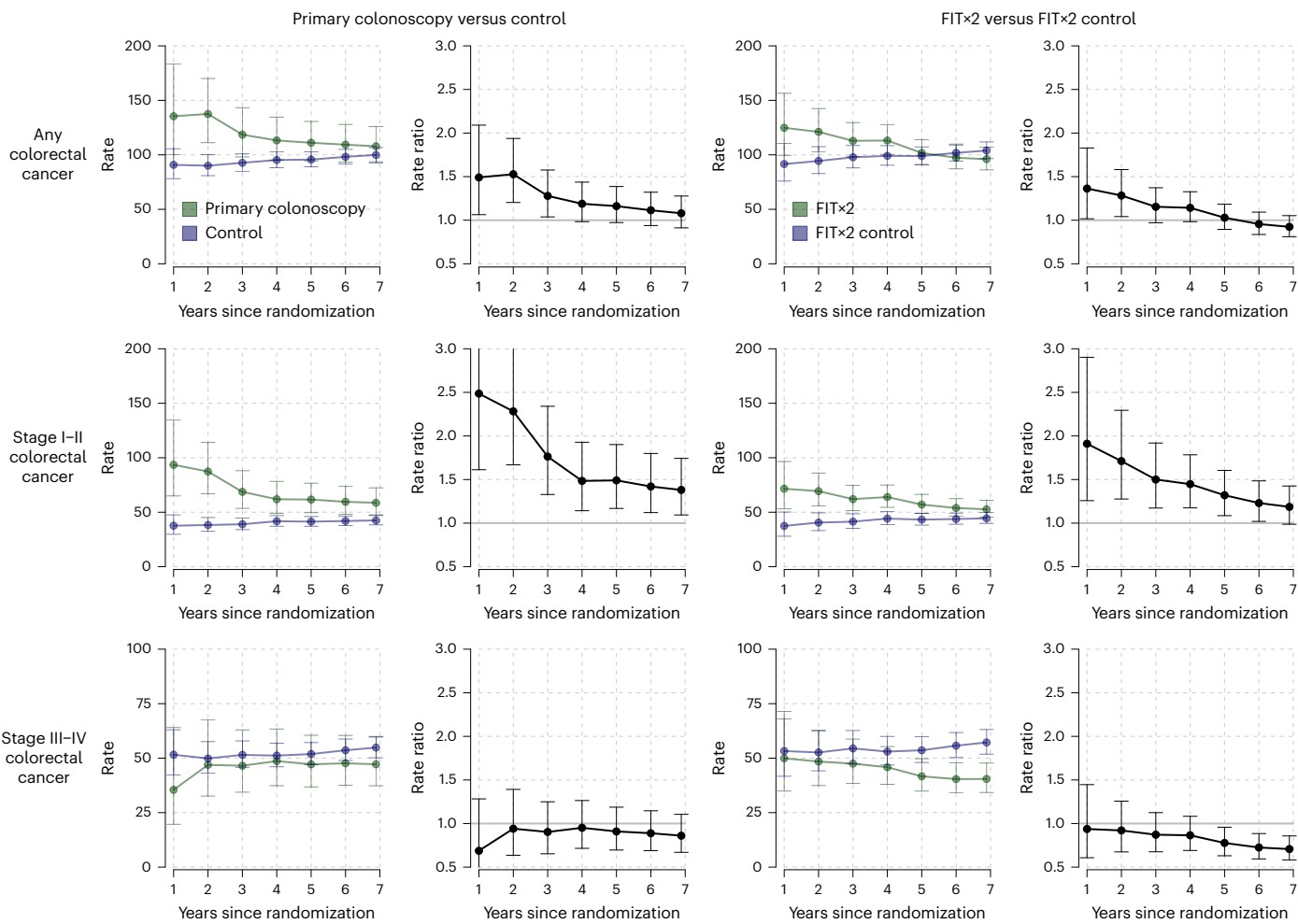

**Fig. 2 | Incidence rate of CRC.** Yearly incidence rate (per 100,000 person-years) and IRRs of CRCs in total and by stage in the primary colonoscopy arm and FIT×2 arm as compared to respective control arms (reference). CRCs include both screen detected and those diagnosed outside of SCREESCO in a usual care setting. Vertical bars indicate 95% CIs.

## Discussion

In this assessment of the diagnostic phase of the SCREESCO trial, there was an increase of stage I–II CRC incidence in the intervention arms compared to usual care, particularly during the first year after randomization when most of the screening colonoscopies were performed. More stage I–II and fewer stage III–IV CRCs were diagnosed in the intervention arms compared to controls after a median follow-up of almost 5 years. Similarly, we found similar incidence rates of death and gastrointestinal or cardiovascular events in all arms, except a slightly higher rate of gastrointestinal and cardiovascular events in particular during the first year after randomization.

The only ongoing RCTs other than the SCREESCO trial, involving both primary colonoscopy and FIT, are the Spanish trial COLONPREV[16] and the American trial Colonoscopy vs. Fecal Immunochemical Test in Reducing Mortality from Colorectal Cancer (CONFIRM)[7,13], neither of which has a control arm. SCREESCO provides a comparison of screening effectiveness between invitation to FIT screening and usual care. In the FIT arms of COLONPREV and CONFIRM, a single stool sample with a higher cutoff was used compared to SCREESCO, which used two stool samples. CONFIRM recruited individuals from Veterans Affairs Medical Centers, where the majority of the population were men[7].

Similar to the Nordic-European Initiative on Colorectal Cancer (NordICC) RCT, we found in intention-to-screen analyses a higher rate of CRC during the first years since randomization to primary colonoscopy compared to usual care, but the rates were similar later

on[9]. Screening using primary sigmoidoscopy has also been shown to reduce CRC incidence in RCTs compared to usual care[18–20] and lead to a stage shift toward lower stage, particularly for screen-detected CRCs[20]. Similar evidence for FIT-based screening compared to usual care is lacking. COLONPREV reported similar rates of screen-detected CRC in the two screening arms[16]. Previous RCTs on screening using repeated fecal occult blood (FOB) followed by colonoscopy after positive FOB test found an initial increase in CRC incidence and a stage shift toward lower stage, especially in screen-detected CRCs[21–24]. In the present study, the rates of stage I–II cancer were higher in both intervention arms compared to controls, in particular during the first years of follow-up when most of the screening colonoscopies were performed. This excess risk decreased with time, and the incidence of stage III–IV simultaneously decreased after around 4 years, particularly in the FIT arm compared to controls. The duration of the trial is likely too short to detect a net benefit of the prevention in terms of a lower CRC incidence in the intervention arms compared to controls and/or to determine if a part of the early-stage CRC excess risk in the intervention arms represents an overdiagnosis of clinically insignificant CRCs[25]. Taken together with the detection and removal of adenomas, our findings suggest a possible future reduction in CRC incidence and CRC mortality in the intervention arms in the subsequent follow-up of SCREESCO (to be reported with follow-up until 31 December 2030)[26].

We previously showed that the rate of serious adverse events directly linked to screening colonoscopies in SCREESCO was low[17] and

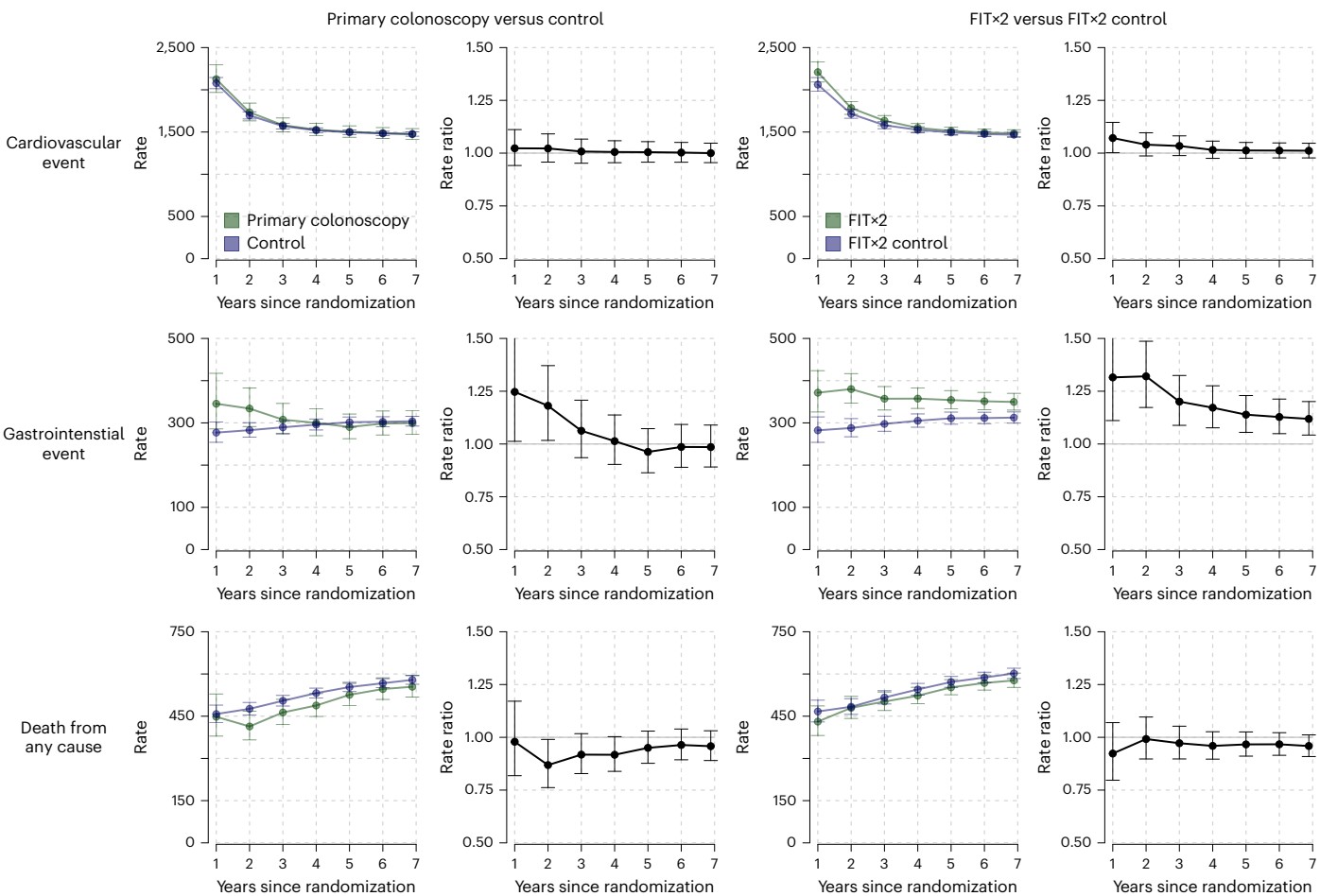

**Fig. 3 | Incidence rate of adverse events.** Yearly incidence rates (per 100,000 person-years) and IRRs of deaths, gastrointestinal events and cardiovascular events in the primary colonoscopy arm and the FIT×2 arm as compared with respective control arms (reference). Vertical bars indicate 95% CIs.

in line with other studies[8,16,27,28]. The present study indicates that the potential harms of colonoscopy in terms of deaths and gastrointestinal or cardiovascular events were generally similar in individuals invited to screening involving colonoscopy at some step in a routine clinical setting and in non-invited individuals following usual care on a population-based level. Individuals who underwent screening colonoscopy in the FIT×2 arm more often had precancerous lesions needing therapeutic intervention compared to the primary colonoscopy arm; hence, the slightly increased risk of gastrointestinal events (that is, bleeding) is expected. The rate of venous thromboembolism was also higher in the FIT×2 arm than in controls, speculatively because anticoagulant users may have a higher risk of positive FIT and pause their treatment during colonoscopy. We are not aware of any other studies reporting gastrointestinal or cardiovascular events in general on an intention-to-treat level comparing primary colonoscopy or FIT followed by colonoscopy to usual care. Our findings motivate further studies on complications related to therapeutic interventions and use of medications.

This large RCT has several strengths. It was performed in a screening-naive setting covering 18 out of 21 of the Swedish regions, comprising 75% of the total population. It had a unique design using two FIT samples instead of one, an unusually low cutoff (10 µg hemoglobin per g) for the maximum of the two FIT values, two rounds of screening 2 years apart and a control arm following usual care. Thirty-three hospitals and 146 endoscopists were involved, which reflects routine care in Sweden. Follow-up was facilitated through reliable and high-quality Swedish registers and allowed us to assess CRCs, gastrointestinal and cardiovascular events and colonoscopies in all arms during the entire

diagnostic phase. The Swedish Cancer Register captures almost all cancers[29], and the proportion of correctly registered diagnoses in the Swedish Patient Register is generally high[30].

This study also has some limitations. Although around half of the invited individuals in the FIT arm participated, which was higher than the COLONPREV study[15], most with a positive FIT participated in colonoscopy. Participation in the primary colonoscopy arm was lower than initially expected but similar to, or higher than, that in many other countries[31] and similar to the NordICC study[8] (outside Norway) and the COLONPREV study[15]. The increase in CRC incidence during the diagnostic phase was likely lower than what would be expected in settings with higher participation in screening. Some gastrointestinal and cardiovascular events may not have been registered in the Swedish Patient Register, and some may have been re-registered at subsequent hospital visits, potentially underestimating or overestimating incidence rates, although we expect the coverage to be similar in all arms and, thus, less likely to influence IRRs. Our study was restricted to individuals aged 60 years and does not inform on the tradeoffs between benefits and harms of the interventions in younger or older individuals[32,33].

In conclusion, we found that a program of screening, by primary colonoscopy or two rounds of two-stool FIT using a low cutoff, detects more lower-stage CRCs than usual care. Rates of gastrointestinal or cardiovascular events were higher in the first year and later on more similar to usual care. The increase in CRC detection implies a benefit of screening while the increase in adverse events suggests some initial harm. Subsequent follow-up of SCREESCO will report CRC mortality (not reported here).

**Table 3 | Incidence rate (per 100,000 person-years) of individuals who died or had a gastrointestinal or cardiovascular event during the diagnostic phase**

| Type of event | Arm | Events | % | Person-years | Rate | 95% CI | IRR[a] | 95% CI |
|---|---|---|---|---|---|---|---|---|
| Cardiovascular event | Primary colonoscopy | 2,111 | 6.8 | 143,037 | 1,475.8 | (1,414.2–1,540.2) | 1.00 | (0.96–1.05) |
| | Control | 12,662 | 6.8 | 857,987 | 1,475.8 | (1,450.3–1,501.7) | 1.00 | |
| | FIT×2 | 4,901 | 8.1 | 330,162 | 1,484.4 | (1,443.4–1,526.6) | 1.01 | (0.98–1.05) |
| | FIT×2 control | 9,688 | 8.0 | 660,123 | 1,467.6 | (1,438.7–1,497.1) | 1.00 | |
| Gastrointestinal event | Primary colonoscopy | 442 | 1.4 | 147,671 | 299.3 | (272.7–328.6) | 0.99 | (0.89–1.09) |
| | Control | 2,689 | 1.4 | 885,246 | 303.8 | (292.5–315.5) | 1.00 | |
| | FIT×2 | 1,193 | 2.0 | 341,297 | 349.5 | (330.3–370.0) | 1.12 | (1.04–1.20) |
| | FIT×2 control | 2,134 | 1.8 | 683,074 | 312.4 | (299.4–326.0) | 1.00 | |
| Death from any cause | Primary colonoscopy | 825 | 2.7 | 148,748 | 554.6 | (518.0–593.8) | 0.96 | (0.89–1.03) |
| | Control | 5,163 | 2.8 | 891,682 | 579.0 | (563.4–595.0) | 1.00 | |
| | FIT×2 | 1,989 | 3.3 | 344,671 | 577.1 | (552.3–603.0) | 0.96 | (0.91–1.01) |
| | FIT×2 control | 4,145 | 3.4 | 688,729 | 601.8 | (583.8–620.4) | 1.00 | |

[a]IRR (rate in intervention arm / rate in control arm).

## Online content

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

**Marcus Westerberg** [1] ✉, **Jonas F. Ludvigsson**[2,3], **Chris Metcalfe**[4], **Ulf Strömberg**[5], **Johannes Blom** [6,7], **Lars Engstrand**[8], **Mikael Hellström** [9,10], **Christian Löwbeer**[11,12], **Robert Steele**[13], **Lars Holmberg**[1,14] & **Anna Forsberg** [15]

[1]Department of Surgical Sciences, Uppsala University, Uppsala, Sweden. [2]Department of Medical Epidemiology and Biostatistics, Karolinska Institutet, Stockholm, Sweden. [3]Department of Pediatrics, Örebro University Hospital, Örebro, Sweden. [4]Population Health Sciences, Bristol Medical School, University of Bristol, Bristol, UK. [5]School of Public Health and Community Medicine, Institute of Medicine, Sahlgrenska Academy at University of Gothenburg, Gothenburg, Sweden. [6]Department of Surgery, Södersjukhuset, Stockholm, Sweden. [7]Department of Clinical Science and Education, Karolinska Institutet, Stockholm, Sweden. [8]Centre for Translational Microbiome Research, Department of Microbiology, Tumor and Cell Biology, Karolinska Institutet, Solna, Sweden. [9]Department of Radiology, Institute of Clinical Sciences at Sahlgrenska Academy, Gothenburg University, Gothenburg, Sweden. [10]Department of Radiology, Sahlgrenska University Hospital, Gothenburg, Sweden. [11]Department of Laboratory Medicine, Division of Clinical Chemistry, Karolinska Institutet, Stockholm, Sweden. [12]Department of Clinical Chemistry, SYNLAB Sverige, Täby, Sweden. [13]Department of Surgery, Population Health and Genomics, School of Medicine, University of Dundee, Ninewells Hospital, Dundee, UK. [14]Translational Oncology & Urology Research (TOUR), School of Cancer and Pharmaceutical Sciences, King's College London, London, UK. [15]Division of Clinical Epidemiology, Department of Medicine K2, Solna, Karolinska Institutet, Stockholm, Sweden. ✉e-mail: marcus.westerberg@uu.se

## Methods

This study has trial number NCT02078804 and is registered with ClinicalTrials.gov at https://clinicaltrials.gov/study/NCT02078804.

### Eligibility criteria

We performed an individual RCT with a study base population from 18 out of 21 regions in Sweden[17] comprising 74.5% of the total national population where CRC screening had not previously been offered (Stockholm, Gotland and Västernorrland regions were not included). Residents aged 60 years of age or who turned 60 in the year of randomization were identified from the Total Population Register maintained by the Swedish Tax Agency[34]. Individuals who had a previous diagnosis of CRC or anal cancer or who had participated in the NordICC trial were excluded[8].

### Consent

All individuals invited for screening signed a written informed consent for the procedure and for biobanking of samples. Individuals assigned as controls were not informed about study participation. The Stockholm Ethics Committee approved the study (2012/2058-31/3) and the review of medical charts (2015/1958-2). The Swedish Ethical Review Authority waived the need for informed consent for accessing pseudonymized register-based data (2022/01946-02 and 2022/06863-2).

### Randomization and masking

Between 11 February 2014 and 1 March 2016, a randomized block method was used to assign 201,000 individuals born 1954–1956 without prior CRC diagnosis to once-only primary colonoscopy, two rounds of FIT 2 years apart (FIT×2) or a usual care control arm with no organized program of screening activity (controls). All controls were intended for use in the separate comparisons of primary colonoscopy versus control and of FIT×2 versus control. Masking was not possible due to the nature of the trial. A list of all eligible individuals within each randomization block, defined by year of randomization, region of residence and sex, was obtained to randomly allocate individuals to the three arms. The target number of randomized individuals within the strata was determined based on the distribution of sex and county among 60-year-old individuals in Sweden (excluding the counties that did not participate in SCREESCO) in 2012. Because of low participation in the primary colonoscopy arm, an additional 77,280 individuals were randomized to primary colonoscopy or control between 30 May 2017 and 25 May 2018. We call the subgroup of controls randomized in 2014–2016 (120,600 controls) 'FIT×2 controls' because they are appropriate for use in the comparison against FIT×2 (also randomized in 2014–2016). All 186,840 controls were available for comparison against primary colonoscopy (randomized 2014–2018).

### Interventions

All invitees in the primary colonoscopy and FIT×2 arms were sent a letter describing the study and a leaflet about CRC and screening. A reminder was sent after 8 weeks. No contact was made with individuals allocated to the control arm. In the primary colonoscopy arm, a second letter offered a scheduled colonoscopy or, if more convenient, a telephone appointment to schedule a colonoscopy. Individuals assigned to the FIT×2 arm were sent a set of kits for two stool samples each screening round. One central laboratory performed all FIT analyses using a single OC-Sensor DIANA automated analyzer (Eiken Chemical). A fecal hemoglobin concentration of ≥10 μg g$^{-1}$ feces in either of the stool samples was deemed positive, triggering a colonoscopy invitation. All individuals in the FIT×2 arm, except those requiring colonoscopy surveillance after adenoma removal or after a CRC diagnosis, were offered a repeat FIT after 2 years, irrespective of participation in the first FIT screening round or the results of the first FIT. Colonoscopies were performed at 33 hospitals by 146 endoscopists, with a background training as gastroenterologists, surgeons or endoscopy nurses[17].

### Diagnostic phase of the trial

Individuals were randomized between 11 February 2014 and 25 May 2018 and subsequently invited to screening. Screening colonoscopies in the primary colonoscopy arm and the FIT×2 arm and FITs in the FIT×2 arm were performed between 2014 and 2020. We, therefore, consider the diagnostic phase of the trial to be between 2014 and 2020.

### Usual care

In Sweden, all citizens have access to public healthcare[35]. A very small minority of individuals have private healthcare insurance on top of this (only 0.6% of Swedish healthcare is funded through insurance). During the study period, there was no national screening. Screening was performed in the Stockholm-Gotland healthcare region but not in the regions where SCREESCO was performed. In usual care, the main driver of colonoscopies is symptoms. During the study period, FIT has been introduced as an intermediate step in the investigation of symptoms to an increasing extent, where elevated hemoglobin in the fecal sample in a FIT taken because of symptoms triggers a colonoscopy. Individuals who are under surveillance due to increased CRC risk (that is, previous CRC diagnosis, inflammatory bowel disease or hereditary/familial CRC syndromes) may undergo colonoscopy during surveillance. Individuals may also be under surveillance after polypectomy of adenomatous colorectal polyps.

### Primary and secondary outcomes of the trial

The ultimate primary endpoint of SCREESCO, for which the power and sample size calculations were performed, is CRC mortality (intervention versus control) at 15 years, and it will be reported later with follow-up until 31 December 2030. CRC incidence was listed as a primary endpoint in the study protocol, but this is formally a secondary outcome of the trial, along with an analysis of compliance, and exploratory outcomes include analyses of health economy, of colonoscopy quality and of the microbiome in feces.

### Summary of changes to the SCREESCO study protocol and statistical analysis plan, 2013–2024

**Changes in the study protocol version 2.0.** The study protocol was amended after new power calculations due to an observed 35% participation in the colonoscopy arm (in Swedish, 10 March 2017; translated to English, 29 April 2021). The list of members of the Scientific Committee was updated.

**Changes in the study protocol version 3.0.** The study protocol was amended after the Scientific Committee decided on a last date of follow-up based on new power calculations. It was also decided that the previously described interim analysis would not be performed. The list of main publications was updated and so was the list of members of the Scientific Committee. A summary of changes to the study protocol and statistical analysis plan was added. Weblinks under 'Head Secretariat' were updated.

**Changes in the statistical analysis plan 2.0.** The statistical analysis plan was amended after new power calculations due to an observed 35% participation in the colonoscopy arm.

**Changes in the statistical analysis plan 3.0.** The statistical analysis plan was amended on 4 November 2024 after the Scientific Committee decided on a last date of follow-up based on new power calculations. It was also decided that the previously described interim analysis would not be performed. Details and clarifications regarding the initial and modified power calculations were added. The analysis of incidence of CRC was changed and is now based on cumulative incidence curves instead of the log-rank test.

## Protocol deviations and rationale for the present study

Power calculations were performed for CRC mortality at 15 years alone. The initial study protocol and the current study protocol state, however, that the main research questions of SCREESCO are to investigate (1) if screening has an effect on the mortality from CRC, (2) if screening has an effect on the incidence of CRC and (3) what method should be used in Sweden regarding the effect according to (1) and (2). The Scientific Committee found a need for a timely assessment of baseline findings (diagnosed CRCs and adverse events during the diagnostic phase of the trial) that also includes the control arm (usual care) and diagnoses/events occurring in general (not only those directly related to SCREESCO screening colonoscopies). The present study is, therefore, listed as a planned main study (section 9, study protocol version 3). Note that CRC mortality is not assessed in the present study and that this main outcome of the trial will, instead, be presented in the final report of the trial with follow-up until 31 December 2030.

Note that this study does not constitute the interim analyses mentioned of CRC mortality at 5 years and 10 years of follow-up in the original study protocol and statistical analysis plan. An additional power calculation (statistical analysis plan version 3; Table 3) showed that there would be limited power to detect differences in CRC mortality at 10 years (follow-up until 31 December 2024) or earlier. Based on the calculation, the Scientific Committee decided that interim analyses of CRC mortality should not be performed.

The present study focuses on early CRCs and adverse events during the diagnostic phase and is listed as a planned analysis of the study. It builds upon a previous baseline assessment of screen-detected CRCs and adverse events in screening colonoscopies of SCREESCO[17] and can be considered an extended baseline report or a 5-year interim analysis of diagnosed CRCs and adverse events occurring during the intervention phase. It includes extended information from Swedish health registers regarding CRCs diagnosed and adverse events in general in both screening arms and in the control arm during the diagnostic phase of SCREESCO when screening FITs and colonoscopies were performed.

## Safety and adverse events

The endoscopy units reported serious adverse events that occurred within 30 days of the colonoscopy. A study nurse, together with A.F., checked the case report forms for completeness or inconsistencies and obtained additional information from the screening units if needed. Subsequent monitoring was, and will be, facilitated by follow-up using healthcare registers.

## Measures of diagnostic yield and adverse events in the present analysis

In the present analysis of the diagnostic phase of SCREESCO, the outcomes were diagnostic yield of screening and usual care, in terms of CRC diagnoses (overall and by stage), and adverse events (cardiovascular and gastrointestinal). We also assessed death from any cause as a measure of overall health and a potential harm of screening.

## Follow-up

In the present study, individuals were followed from randomization (2014–2018) until positive outcome in the corresponding analysis (CRC diagnosis or adverse event, respectively), emigration, death or end of the diagnostic phase (31 December 2020), whichever occurred first, through linkage between the Cancer Register, the Patient Register, the Total Population Register, the Cause of Death Register and the SCREESCO database.

Data on date of and stage at CRC diagnosis were extracted from the Swedish Cancer Register, to which registration is mandated by law[29]. Additional data on CRC stage were obtained from the Swedish Colorectal Cancer Register that, during 2008–2015, had a completeness of 98.5% for colon cancer and 98.8% for rectal cancer[36]. We defined CRC according to relevant International Classification of Disease (ICD) and Systematized Nomenclature of Medicine (SNOMED) codes registered in the Swedish Cancer Register. We used any of the following ICD 10th revision (ICD-10) codes for CRC: C18 (except C181), C19 and C20. We also required at least one of the following SNOMED version 3 (SNOMED/3; morphology code ICD-0/3) codes registered at the same time as the CRC: 81403 (adenocarcinoma), 82113 (tubular adenocarcinoma), 82133 (serrated adenocarcinoma), 82203 (adenocarcinoma in familial polyposis), 82433 (goblet cell adenocarcinoma), 82613 (adenocarcinoma arising from villous adenoma), 82633 (adenocarcinoma arising from tubulovillous adenoma), 84803 (mucinous adenocarcinoma) and 84903 (signet ring cell carcinoma/poorly cohesive carcinoma); or SNOMED/2 (SNOMEDO10; morphology code ICD-0/2): 81403 (tubular, villous or serrated adenocarcinoma), 82113 (tubular adenocarcinoma), 82203 (goblet cell adenocarcinoma or adenocarcinoma in familial polyposis), 84803 (goblet cell or mucinous adenocarcinoma) and 84903 (signet ring cell carcinoma/poorly cohesive carcinoma). We additionally included CRCs registered in the Swedish Colorectal Cancer Quality Register[36].

The date of CRC diagnosis was defined as the first date of a CRC diagnosis in either of the two registries. Medical charts for individuals with CRC registered in SCREESCO, individuals who after a screening colonoscopy required further investigation or treatment (for example, computed tomography scan) and individuals whose polyp samples were sent to pathology were reviewed to assess the correctness of the CRC diagnosis and to extract additional information on stage. CRCs were considered screen detected if detected at colonoscopy (primary or secondary after a positive FIT) performed within SCREESCO.

Data on stage of CRCs detected at screening colonoscopy within SCREESCO were extracted from the SCREESCO database. Data on stage were also extracted separately for all individuals with a CRC diagnosis in the Swedish Colorectal Cancer Quality Register and/or the Swedish Cancer Register. We allowed for some administrative lag and included all entries in the Swedish Colorectal Cancer Quality Register and the Swedish Cancer Register occurring within 90 days from the first date of CRC diagnosis in either of the two registers. The TNM stages were obtained from the Swedish Colorectal Cancer Quality Register if the CRC was registered there and available (not missing) and, otherwise, from the Swedish Cancer Register if available. The Swedish Colorectal Cancer Quality Register contains both the clinical and pathological TNM, and the pathological TNM was used if available and, otherwise, the clinical TNM. Stage (I–II versus III–IV) was graded similarly to the American Joint Committee on Cancer system, which is based on the TNM classification, with stage I–II if T(any or missing)N0M0, T(0–4) N0M(0/missing) or T(0–1)N(0/missing)M0 and stage III–IV if N1, N2 or M1. Tx was considered as missing T stage and similarly for N stage and M stage. Stage was considered unknown if it did not fulfil any of the above.

Data on cardiovascular and gastrointestinal events (events related to colonoscopy – for example, bleedings and injuries) and on colonoscopies registered in an inpatient or outpatient setting were extracted from the Patient Register[30]. The Patient Register covers all inpatient care in Sweden, and the validity of this register has been found to be high, varying from 85% to 95% among different diseases[30]. The following ICD-10 codes were used to define gastrointestinal events: K922 (Unspecified gastrointestinal bleeding), S360 (Splenic injury), S365 (Colonic injury), S366 (Rectal injury), T810 (Bleeding iatrogenic) and T812 (Perforation). The following ICD-10 codes were used to define cardiovascular events: I20–I25 (Ischemic heart disease), I26 (Pulmonary embolism), I33 (Acute endocarditis), I46 (Cardiac arrest), I63 (Cerebral infarction), I74 (Peripheral artery embolism) and I81 and I82 (Venous thromboembolism).

Date of death was retrieved from the Cause of Death Register[37].

## Covariates

Biological sex (man/woman), year of birth, country of birth (Sweden versus other), region of residence, educational level, Charlson

Comorbidity Index, a drug comorbidity index and history of any cardiovascular or gastrointestinal event and type of event (within 10 years prior to randomization) were used to describe individuals by arm. The Charlson Comorbidity Index was calculated based on the Patient Register in the last 10 years before inclusion, and the drug comorbidity index was calculated based on drug prescriptions registered up to 1 year before date of inclusion in the Prescribed Drug Register[38–40]. Country of birth was extracted from the Total Population Register, and educational level and region of residence were extracted from the Swedish Longitudinal Integrated Database for Health Insurance and Labor Market Studies (LISA).

### Power calculations

The sample size was calculated based on the primary endpoint of CRC mortality. A priori, the study was powered to detect a 17.5% decrease in CRC mortality at 15 years in individuals invited to colonoscopy compared to the control arm and a 15% decrease in individuals invited to FIT compared to the control arm, based on an anticipated 1% cumulative CRC mortality between 60 years and 75 years of age. The original sample size target was 201,000 individuals.

Because of lower participation (35%) in the primary colonoscopy arm than expected (50%), new power calculations were performed to determine the additional number of randomized individuals needed to achieve acceptable power, and two additional age cohorts (born 1957 and 1958) were randomized to primary colonoscopy or control in a ratio of 1:6 (hence the deviation from the initially intended allocation ratio).

In the revised study protocol, we assumed a 15% disease-specific mortality reduction by 15 years of follow-up as a minimal clinically important effect in those invited to FIT×2 compared to the control arm, based on a participation rate of 50%, and a 17.5% disease-specific mortality reduction in those invited to colonoscopy compared to the control arm, based on a 35% participation. To allow these absolute risk reductions to be detected at a two-sided 2.5% significance level with 80% power for the comparison of FIT×2 versus control and 73% power for the comparison of primary colonoscopy versus control, the revised target sample size was 278,280 participants. The significance level was adjusted for two comparisons according to the Bonferroni method. In total, 31,140 individuals were randomized to the primary colonoscopy arm; 60,300 individuals were randomized to the FIT×2 arm; and there were two control groups: 186,840 controls to the primary colonoscopy arm, out of which 120,600 individuals also were controls to the FIT×2 arm (FIT×2 controls).

An additional power analysis was performed to determine when, in calendar time, the main analysis of SCREESCO would be performed because this was not clearly stated in the initial study protocol. The Scientific Committee decided that 31 December 2030 would be the last date of follow-up for the main analysis because power was not expected to meaningfully increase after this date.

### Statistical analysis

In an intention-to-screen analysis, we report the number, proportion and incidence rate per 100,000 person-years of all screen-detected CRCs within SCREESCO and all other CRCs diagnosed in regular clinical practice during the diagnostic phase of SCREESCO, in total and by stage (I–II or III–IV). We similarly report incident cardiovascular and gastrointestinal events diagnosed in regular clinical practice and death from any cause. We compare each intervention arm with the corresponding control arm concerning each of the outcomes using incidence rates and IRRs using Poisson regression models with 95% CIs. Analyses were also performed separately in men and women, and a Poisson regression model with interaction between sex and study arm was used to compare the IRR in men versus the IRR in women under the null hypothesis of no difference (two-sided test of the interaction term). Note that only the FIT×2 controls (individuals randomized 2014–2016, during which individuals could be allocated to FIT×2) were used in the comparison

of FIT×2 versus control in this study, whereas all controls (randomized 2014–2018, including the FIT×2 controls) were used in the comparison of primary colonoscopy versus control.

The full follow-up of up to a maximum of almost 7 years was used in the above analyses. In a complementary analysis, we also computed incidence rates and IRRs at each year of follow-up. Individuals were censored if and when they migrated out of Sweden. Individuals were similarly not considered at risk after their date of death. We assessed potential violations of equidispersion for each regression model and computed alternative CIs by use of robust standard errors. Because there were no signs of meaningful underdispersion or overdispersion and results were virtually identical, we did not report these analyses.

In a complementary analysis, we computed competing risk cumulative incidence curves of CRC (overall and by stage) where death from any cause was considered a competing risk.

We report baseline characteristics in participants and non-participants of the intervention arms along with diagnosed CRCs and cardiovascular and gastrointestinal events that occurred during the intervention phase. Individuals in the primary colonoscopy arm were considered to be participants if they underwent a screening colonoscopy, and individuals in the FIT×2 arm were considered participants if they returned a FIT in any of the two rounds.

Stata version 13.1 and R version 4.0.2 were used for power calculations. Analyses were performed using R version 4.0.2.

### Reporting summary

Further information on research design is available in the Nature Portfolio Reporting Summary linked to this article.

### Data availability

The data cannot be shared publicly because the individual-level data contain potentially identifying and sensitive patient information and cannot be published due to legislation and ethical approval (https://etikprovningsmyndigheten.se). Use of the data from national health data registers is further restricted by the Swedish Board of Health and Welfare (https://www.socialstyrelsen.se/en/) and Statistics Sweden (https://www.scb.se/en/), which are government agencies providing access to the linked healthcare registers. Selected deidentified individual participant data that underlie the results reported in this article (including in the supplementary materials) can, however, be made available to researchers after request to the SCREESCO Steering Committee. Researchers must provide a methodologically sound proposal for a project that conforms with the Swedish Ethical Review Authority permit for the project and will need to sign a data access agreement. Data will be made available at a secure remote server to achieve the aims in the approved proposal. Data will be available from 3 months after publication and until 3 years after publication of the article. Proposals regarding the data underlying this article may be submitted up to 2 years after publication. The SCREESCO study will not carry the costs of external projects. The full trial protocol and statistical analysis plan (including original and revised versions) are available in Supplementary Note 1, and the most recent version of each document is available at https://clinicaltrials.gov/study/NCT02078804.

### Code availability

The code was written in R version 4.0.2 using RStudio and is available at https://github.com/MarcusWesterberg/CRCs-and-AEs-during-diagnostic-phase-of-SCREESCO.

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

## Acknowledgements

We are grateful for grants from the 18 Swedish regions, the Stockholm County Council, Regional Cancer Center Mellansverige, the Swedish Cancer Society, the Aleris Research and Development Fund and Eiken Chemical. We are also grateful to the 33 Swedish hospitals where the colonoscopies in this study were performed. Late Rolf Hultcrantz is acknowledged for initiating SCREESCO, securing the funding and leading the study through the entire recruitment period and first report. Late Anders Ekbom is acknowledged for his important contributions in all phases of the study. Financial support was provided by the Swedish regions (A.F.), Regional Cancer Center Mellansverige (A.F.), the Swedish Cancer Society (A.F.; 2018/595 and 2021/1628), the Aleris Research and Development Fund (A.F.) and Eiken Chemical (A.F.). Financial support was also provided through the Regional Agreement on Medical Training and Clinical Research (A.L.F.) between the Stockholm County Council and the Karolinska Institutet (A.F.) and by the Swedish Society of Medicine (A.F.; Ihre Foundation, SLS-961166). The funders of the study had no role in study design, data collection, data analysis, data interpretation or writing of the report.

## Author contributions

M.W. and A.F. directly accessed and verified the underlying data reported in the paper. Conceptualization: all authors. Data curation: M.W. and A.F. Formal analysis: M.W. Funding acquisition: A.F. Investigation: all authors. Methodology: M.W., J.F.L., C.M., L.H. and A.F. Project administration: J.F.L. and A.F. Resources: A.F. Software: M.W. Supervision: A.F. Validation: M.W. and A.F. Visualization: M.W. Writing—original draft: M.W., J.F.L., L.H. and A.F. Writing—review and editing: all authors.

## Funding

## Competing interests

J.F.L. has coordinated an unrelated study on behalf of the Swedish IBD quality register (SWIBREG). That study received funding from Janssen corporation. J.F.L. has also received financial support from Merck Sharp & Dohme/Merck for developing a paper reviewing national healthcare registers and has ongoing research collaboration on inflammatory bowel disease. J.F.L. has an ongoing research collaboration on celiac disease with Takeda Pharmaceuticals and has ongoing discussions for future research collaboration on liver disease with this company. The other authors declare no competing interests.

## Additional information

**Extended data** is available for this paper at https://doi.org/10.1038/s41591-026-04225-9.

**Correspondence and requests for materials** should be addressed to Marcus Westerberg.

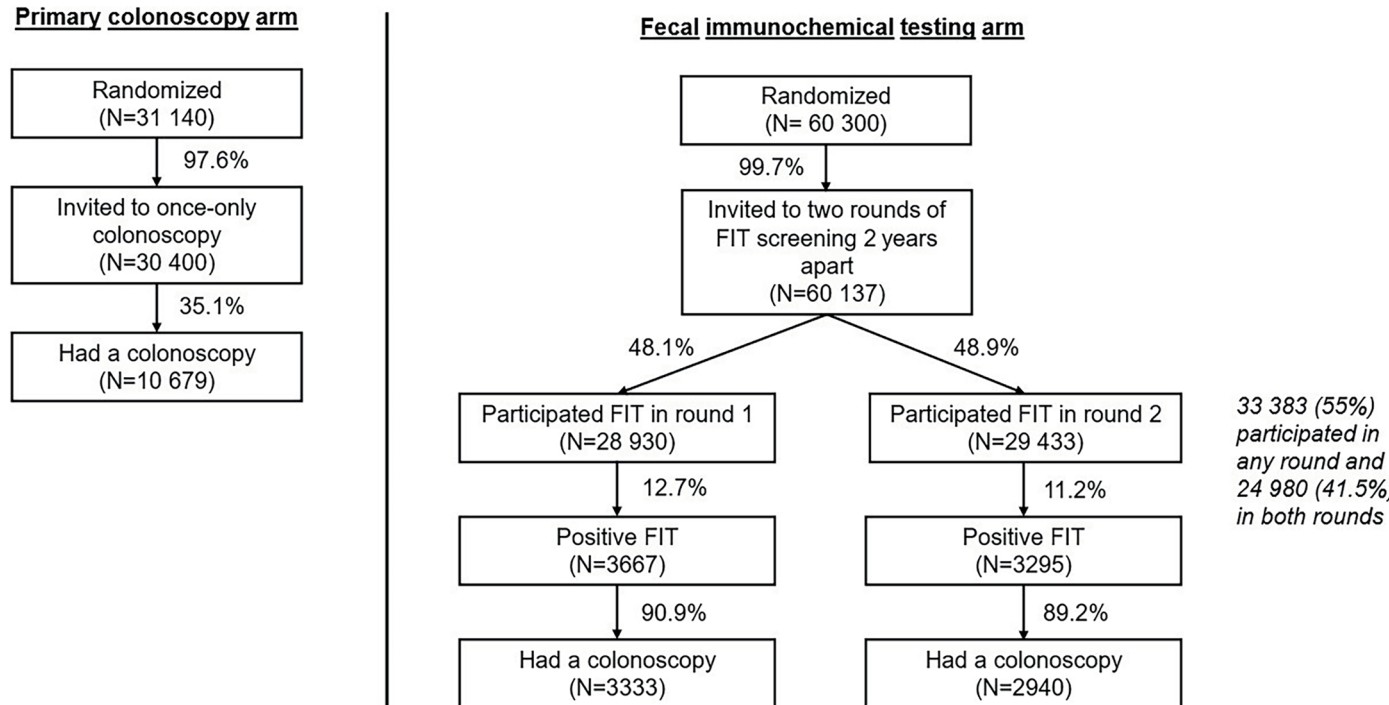

**Extended Data Fig. 1 | Participation.** Number of invited and participating individuals according to study arm.

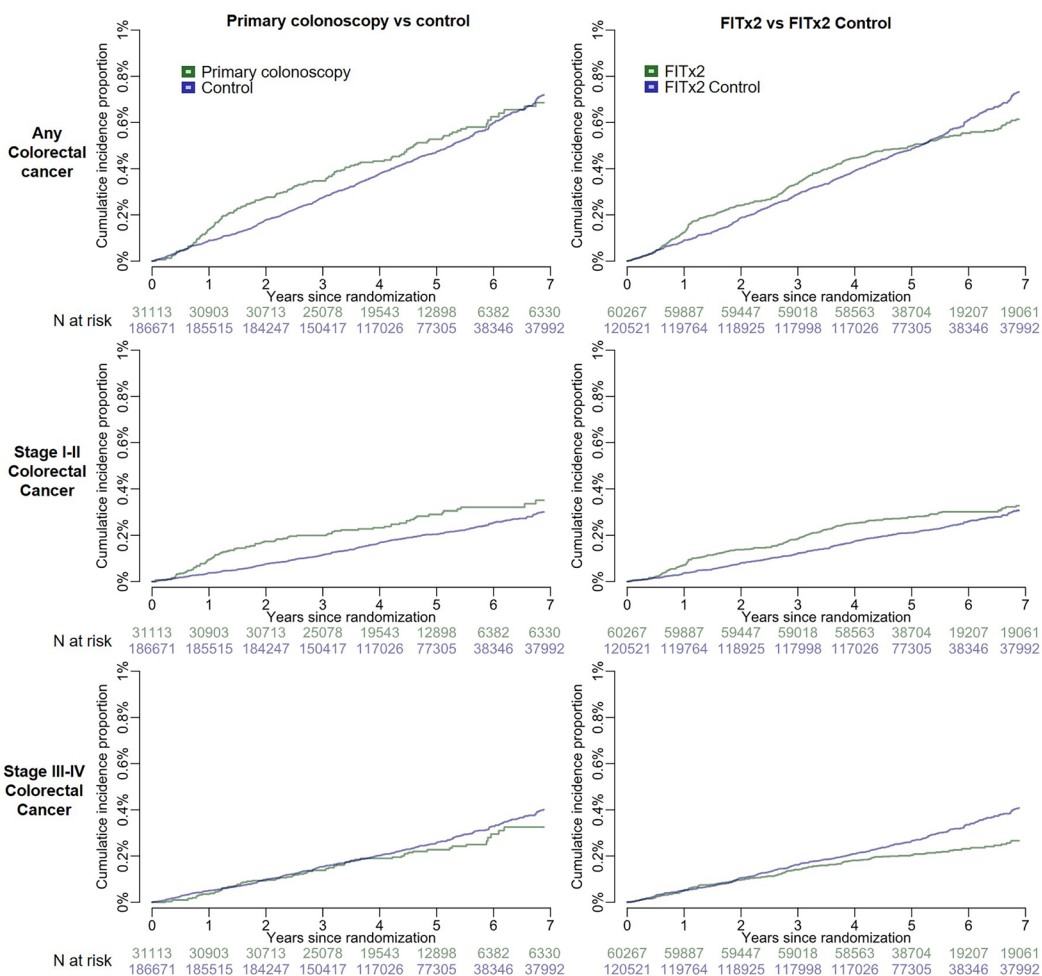

**Extended Data Fig. 2 | Colorectal cancer incidence.** Completing risk cumulative incidence proportion of colorectal cancer with death from any cause as completing risk.

# Reporting Summary

## Statistics

For all statistical analyses, confirm that the following items are present in the figure legend, table legend, main text, or Methods section.

| n/a | Confirmed | |
|---|---|---|
| ☐ | ☒ | The exact sample size (*n*) for each experimental group/condition, given as a discrete number and unit of measurement |
| ☒ | ☐ | A statement on whether measurements were taken from distinct samples or whether the same sample was measured repeatedly |
| ☒ | ☐ | The statistical test(s) used AND whether they are one- or two-sided<br>*Only common tests should be described solely by name; describe more complex techniques in the Methods section.* |
| ☐ | ☒ | A description of all covariates tested |
| ☐ | ☐ | A description of any assumptions or corrections, such as tests of normality and adjustment for multiple comparisons |
| ☐ | ☒ | A full description of the statistical parameters including central tendency (e.g. means) or other basic estimates (e.g. regression coefficient) AND variation (e.g. standard deviation) or associated estimates of uncertainty (e.g. confidence intervals) |
| ☒ | ☐ | For null hypothesis testing, the test statistic (e.g. *F*, *t*, *r*) with confidence intervals, effect sizes, degrees of freedom and *P* value noted<br>*Give P values as exact values whenever suitable.* |
| ☒ | ☐ | For Bayesian analysis, information on the choice of priors and Markov chain Monte Carlo settings |
| ☒ | ☐ | For hierarchical and complex designs, identification of the appropriate level for tests and full reporting of outcomes |
| ☐ | ☒ | Estimates of effect sizes (e.g. Cohen's *d*, Pearson's *r*), indicating how they were calculated |

*Our web collection on statistics for biologists contains articles on many of the points above.*

## Software and code

Policy information about availability of computer code

| Data collection | Stata (version 13.1) and R version 4.0.2 were used for power calculations. |
|---|---|
| Data analysis | Analyses were performed using R version 4.0.2. Code is available at https://github.com/MarcusWesterberg/CRCs-and-AEs-during-diagnostic-phase-of-SCREESCO |

For manuscripts utilizing custom algorithms or software that are central to the research but not yet described in published literature, software must be made available to editors and reviewers. We strongly encourage code deposition in a community repository (e.g. GitHub). See the Nature Portfolio guidelines for submitting code & software for further information.

## Data

Policy information about availability of data

All manuscripts must include a data availability statement. This statement should provide the following information, where applicable:
- Accession codes, unique identifiers, or web links for publicly available datasets
- A description of any restrictions on data availability
- For clinical datasets or third party data, please ensure that the statement adheres to our policy

The data cannot be shared publicly because the individual-level data contain potentially identifying and sensitive patient information and cannot be published due to legislation and ethical approval (https://etikprovningsmyndigheten.se). Use of the data from national health-data registers is further restricted by the Swedish Board of Health and Welfare (https://www.socialstyrelsen.se/en/) and Statistics Sweden (https://www.scb.se/en/) which are Government Agencies providing access

to the linked healthcare registers. Selected deidentified individual participant data that underlie the results reported in this Article (including in the supplement), can however be made available to researchers after request to the SCREESCO Steering Committee. Researchers must provide a methodologically sound proposal for a project that conforms with the Swedish Ethical Review Authority permit for the project and will need to sign a data access agreement. Data will be made available at a secure remote server to achieve the aims in the approved proposal. Data will be available from 3 months after publication and until 3 years after publication of the Article. Proposals regarding the data underlying this Article may be submitted up to 2 years after publication. The SCREESCO study will not carry the costs of external projects. The full trial protocol and statistical analysis plan (including original and revised versions) are available in Supplementary Information and the most recent version of each document is available at https://clinicaltrials.gov/study/NCT02078804.

# Research involving human participants, their data, or biological material

Policy information about studies with human participants or human data. See also policy information about sex, gender (identity/presentation), and sexual orientation and race, ethnicity and racism.

| Reporting on sex and gender | Sex-based analyses are included |
| --- | --- |
| Reporting on race, ethnicity, or other socially relevant groupings | Country of birth (Sweden and other) is reported in Table 1 |
| Population characteristics | Sex, year of randomization, health care region, educational level, comorbidity |
| Recruitment | We did a randomized controlled trial with a study base population from 18 out of 21 regions in Sweden comprising 74.5% of the total national population where CRC screening had not previously been offered (Stockholm, Gotland, and Västernorrland were not included). Residents aged 60 years in the year of randomization (2014-2018) were identified through the Swedish Total Population Register. Individuals who had a previous diagnosis of CRC or anal cancer or who had participated in the ongoing NordICC trial were excluded. |
| Ethics oversight | The Stockholm Ethics Committee approved the study (2012/2058-31/3) and the review of medical charts (2015/1958-2). The Swedish Ethical Review Authority waived the need for informed consent for accessing pseudonymised register-based data (2022/01946-02 and 2022/06863-2). |

Note that full information on the approval of the study protocol must also be provided in the manuscript.

# Field-specific reporting

Please select the one below that is the best fit for your research. If you are not sure, read the appropriate sections before making your selection.

☒ Life sciences ☐ Behavioural & social sciences ☐ Ecological, evolutionary & environmental sciences

For a reference copy of the document with all sections, see nature.com/documents/nr-reporting-summary-flat.pdf

# Life sciences study design

All studies must disclose on these points even when the disclosure is negative.

| Sample size | A randomized block method was used to assign individuals born 1954-1956 without prior CRC diagnosis to once-only primary colonoscopy, two rounds of fecal immunochemical testing 2 years apart (FITx2), or a usual care control arm with no organised program of screening activity (controls). Masking was not possible due to the nature of the trial.<br><br>The sample size was calculated using the STATA function "stpower" on the basis of the primary endpoint of colorectal cancer mortality. The original sample size target was 201 000 individuals, based on an assumed 1% cumulative colorectal cancer mortality for a follow-up from age 60 to 75 years. Because of lower participation (35%) in the primary colonoscopy arm than expected (50%), two additional age cohorts (born 1957 and 1958) were randomized to primary colonoscopy or control. New power calculations were performed based on the lower participation rate in the primary colonoscopy arm to determine the additional number of randomized individuals.<br><br>In the revised study plan, we assumed a 15% disease-specific mortality reduction by 15 years of follow-up as a minimal clinically important effect in those invited to FITx2, based on a participation rate of 50%, and a 17.5% disease-specific mortality reduction in those invited to colonoscopy, based on a 35% participation rate. To allow differences to be detected at a two-sided 2.5% significance level using the log rank test with 80% power for the comparison of FITx2 versus control and 73% power for the comparison of primary colonoscopy versus control the revised target sample size was 278 280 participants. The significance level was adjusted for two comparisons according to the Bonferroni method. In total, 31 140 individuals were randomized to the primary colonoscopy arm, 60 300 to FITx2, and two control groups: 186 840 controls to the primary colonoscopy arm out of which 120 600 individuals also were controls to the FITx2 arm (FITx2 controls). |
| --- | --- |
| Data exclusions | Between February 11, 2014, and May 25, 2018, 278 280 individuals were randomized. Due to administrative lag in registration, 159 dead and 70 with prevalent CRC before the date of randomization were identified only after randomization and were excluded. Another two control individuals were excluded since they were not identifiable in any register. |
| Replication | RCT of individuals randomized to primary colonposcopy, FIT or usual care, so not replicable |
| Randomization | A randomized block method was used to assign individuals without prior CRC diagnosis to once-only primary colonoscopy, two rounds of fecal immunochemical testing 2 years apart (FITx2), or a usual care control arm with no organised program of screening activity (controls). |

We have not identified any reasons for self-selection bias or other biases in the randomization process.

Blinding | Masking/blinding was not possible due to the nature of the trial.

# Reporting for specific materials, systems and methods

We require information from authors about some types of materials, experimental systems and methods used in many studies. Here, indicate whether each material, system or method listed is relevant to your study. If you are not sure if a list item applies to your research, read the appropriate section before selecting a response.

## Materials & experimental systems

| n/a | Involved in the study |
|-----|-----------------------|
| ☒ ☐ | Antibodies |
| ☒ ☐ | Eukaryotic cell lines |
| ☒ ☐ | Palaeontology and archaeology |
| ☒ ☐ | Animals and other organisms |
| ☐ ☒ | Clinical data |
| ☒ ☐ | Dual use research of concern |
| ☒ ☐ | Plants |

## Methods

| n/a | Involved in the study |
|-----|-----------------------|
| ☒ ☐ | ChIP-seq |
| ☒ ☐ | Flow cytometry |
| ☒ ☐ | MRI-based neuroimaging |

## Clinical data

Policy information about clinical studies

All manuscripts should comply with the ICMJE guidelines for publication of clinical research and a completed CONSORT checklist must be included with all submissions.

Clinical trial registration | ClinicalTrials.gov NCT02078804

Study protocol | https://clinicaltrials.gov/study/NCT02078804 and in Supplementary Information

Data collection | We did a randomized controlled trial with a study base population from 18 out of 21 regions in Sweden comprising 74.5% of the total national population where CRC screening had not previously been offered (Stockholm, Gotland, and Västernorrland were not included). Residents aged 60 years in the year of randomization (2014-2018) were identified through the Swedish Total Population Register and randomly allocated.

Outcomes | The primary outcome of the trial is colorectal cancer mortality and secondary outcomes include incidence of colorectal cancer and adverse events, see the most recent version of the Study Protocol. In the current analysis we analyzed these secondary outcomes during the diagnostic phase of the trial, i.e. diagnostic yield of screening and usual care, in terms of CRC diagnoses (overall and by stage), and adverse events (cardiovascular and gastrointestinal and death from any cause). These were ascertained using national data registries and assessed using incidence rates and incidence rate ratios for comparison between intervention and control.

## Plants

Seed stocks | *Report on the source of all seed stocks or other plant material used. If applicable, state the seed stock centre and catalogue number. If plant specimens were collected from the field, describe the collection location, date and sampling procedures.*

Novel plant genotypes | *Describe the methods by which all novel plant genotypes were produced. This includes those generated by transgenic approaches, gene editing, chemical/radiation-based mutagenesis and hybridization. For transgenic lines, describe the transformation method, the number of independent lines analyzed and the generation upon which experiments were performed. For gene-edited lines, describe the editor used, the endogenous sequence targeted for editing, the targeting guide RNA sequence (if applicable) and how the editor was applied.*

Authentication | *Describe any authentication procedures for each seed stock used or novel genotype generated. Describe any experiments used to assess the effect of a mutation and, where applicable, how potential secondary effects (e.g. second site T-DNA insertions, mosiacism, off-target gene editing) were examined.*

