## [Peer Review File · Nature Medicine]

Colonoscopy and fecal immunochemical testing vs usual care in diagnostic colorectal cancer screening: the SCREESCO randomized controlled trial

Corresponding Author: Mr Marcus Westerberg

Version 1:

Reviewer comments:

Reviewer #1

(Remarks to the Author)

Results from the colorectal cancer screening trial SCREESCO diagnostic phase (end of follow-up, December 31, 2020) are presented. While the authors indicate that this is a prespecified analysis, no corresponding details could be found in the protocol. It is unclear whether using a cutoff at 2020, with a median follow-up of 4.8 years, provides a meaningful assessment of incidence endpoints. Longer follow-up may be more informative, especially given that nearly five years have passed since 2020.

Could the authors comment on potential overdiagnosis, given the elevated number of cancers detected in the colonoscopy arm? Additionally, how are deaths handled when calculating the incidence rates reported in the manuscript?

It may also be valuable to report per-protocol results for comparison. Furthermore, what proportion of participants in the FIT arm actually underwent colonoscopy following a positive FIT test? How are these participants classified in terms of compliance, and if they are not considered noncompliant, could the authors clarify why?

Reviewer #2

(Remarks to the Author)

Thank you for the opportunity to review this important manuscript based on the SCREESCO trial. This is a large-scale randomized controlled study addressing a key question in population-level colorectal cancer (CRC) screening. However, I have several major concerns regarding the current version of the manuscript, which may limit the interpretability, internal validity, and practical implications of the findings.

1) The manuscript lacks detailed reporting of participation rates in each screening arm and for each round of FIT screening. Given that screening effectiveness is highly sensitive to uptake, these details are essential for evaluating both internal validity and external generalizability. Some of this information has been previously published (e.g., *Lancet Gastroenterol Hepatol.* 2022;7:513–521), but should be explicitly referenced and integrated into the current manuscript.

2) Only intention-to-treat (ITT) analyses are presented. Considering the low participation rates reported for both colonoscopy and FIT, ITT analyses alone are insufficient to assess the true effectiveness of the interventions. Per-protocol and as-screened analyses would provide essential complementary information on the potential benefit among participants who actually underwent screening.

3) The rationale for using a shared control group across the two screening comparisons (colonoscopy vs usual care and FIT vs usual care) is not clearly explained. This design choice raises questions about comparability and potential statistical dependencies, and should be explicitly justified.

4) The manuscript focuses on CRC incidence and mortality, but does not report data on advanced adenomas, which are critical intermediate outcomes. Including this information would enhance understanding of the potential for long-term CRC prevention, particularly given the relatively short follow-up time.

5) The median follow-up period of 4.8 years is likely insufficient to capture the full preventive effect of screening, particularly for CRC mortality. A longer follow-up (e.g., 10 years) is typically necessary to assess the true impact of screening. The addition of Kaplan–Meier curves for cumulative incidence and mortality would also improve the clarity and transparency of outcome reporting.

6) While the manuscript concludes that screening increased detection of early-stage CRC, it does not clearly address the finding that neither screening strategy reduced CRC incidence or mortality within the observed timeframe. Given that incidence was the primary outcome, this null result should be more prominently presented and discussed in the conclusion section.

7) The limited participation rates and the restriction of the study population to individuals aged 60 years and above should be acknowledged as key limitations. These factors substantially affect the generalizability of the findings and limit their applicability to broader screening populations or policy settings.

Reviewer #3

(Remarks to the Author)

In this study, the authors report on the initial diagnostic phase of the SCREESCO trial. This is a large, population-based, randomized controlled trial in Sweden, which was designed to assess the effectiveness of two different colorectal cancer (CRC) screening strategies compared to usual care. In this study, the authors conducted a pre-specified analysis of the trial's diagnostic phase to assess early outcomes. The outcomes for this analysis were the incidence of CRC (overall and by stage), all-cause mortality, and the incidence of gastrointestinal and cardiovascular adverse events. Overall, the study addresses an important public health question, and the manuscript is generally well-written. However, there are several issues that require attention.

1. The most recent Statistical Analysis Plan (version 3.0, 2025-01-16) explicitly states, "The analysis of incidence of colorectal cancer was changed and is now based on cumulative incidence curves instead of the log-rank test." It is not clear whether it is also designed for this prespecified analysis since the details for statistical analyses of this study is not fully described by the protocol. The current manuscript focuses on CRC incidence whereas it does not present cumulative incidence curves. Instead, it reports incidence rates and uses Poisson regression to calculate rate ratios. The authors should address this discrepancy, or they should perform the analysis as specified in the Statistical Analysis Plan (i.e., present cumulative incidence curves and associated statistical tests) and update the manuscript accordingly.

2. The protocol and Statistical Analysis Plan (version 3.0) state, "It was also decided that the previously described interim analysis will not be performed". In Appendix B, they also state "The scientific committee found a need for a timely assessment of baseline findings (diagnosed CRCs and adverse events during the diagnostic phase of the trial) that also includes the control arm (usual care) and diagnoses/events occurring in general (not only those directly related to SCREESCO screening colonoscopies)". Why is the interim analysis canceled, and what is the need for performing this analysis given that the study is powered enough? These should be justified. Also, the authors should clarify in the main manuscript's Methods section that this analysis is not the formal interim analysis that was cancelled. They should clearly state the purpose for this pre-specified analysis to avoid any confusion.

3. While the exclusions of participants post-randomization based on registry linkage are clearly reported (e.g., deaths or CRC diagnoses prior to randomization), it is unclear whether there was any additional loss to follow-up (e.g., due to migration, data linkage failure, or censoring not at death or event). The authors should explicitly report the number of participants lost to follow-up and provide a brief description of censoring mechanisms.

4. Sensitivity analyses should be added. For instance, overdispersion should be assessed for the Poisson models given the large, heterogeneous trial population with regional and clinical variability. If overdispersion is present, the authors may consider addressing it using quasi-Poisson, negative binomial models, or robust standard errors to ensure valid inference.

5. In the main text, it would be helpful to briefly cover what constitutes "usual care" for CRC screening in Sweden for this age group, describing the usual care pathway (e.g., primarily symptomatic detection, availability of opportunistic screening).

6. I would suggest the authors add explanations for the statistical power of this study because there are several times of adjustments on the sample size due to the lower-than-expectation compliance. For example, in the main text, it was mentioned that the expected statistical power for FITx2 was 80% and 73% for direct colonoscopy. This is somehow weird because the expected effect size for direct colonoscopy should be larger than FIT, though the most recent version of Statistical Analysis Plan has clarified on this.

Version 2:

Reviewer comments:

Reviewer #1

(Remarks to the Author)

Histologic confirmation rules out false positives but does not eliminate the possibility of overdiagnosis, as some pathologically confirmed cancers may still be slow-growing and clinically insignificant. The increased number of early-stage cancer diagnoses in the intervention arm could therefore reflect overdiagnosis. Particularly if this rise is not accompanied by a corresponding decline in late-stage or metastatic cancers and cancer-related deaths, which is the hallmark of overdiagnosis. A reduction in late-stage cancer incidence may serve as a more meaningful intermediate outcome. However,

it is unclear whether the current diagnostic-phase results, based on a median follow-up of 4.8 years, are sufficient to yield meaningful conclusions.

Reviewer #2

(Remarks to the Author)

The SCREESCO study indeed represents an important randomized controlled trial (RCT) in the field of colorectal cancer (CRC) screening, and the authors have made efforts to address several points raised by the reviewers. However, despite these revisions, the current manuscript provides only limited additional evidence beyond what is already available from recent large-scale RCTs.

Two major RCTs on CRC screening have been published recently — the COLOPREV trial (Lancet. 2025;405:1231–1239) and the NORDICC trial (N Engl J Med. 2022;387:1547–1556). Both provide robust data on the effects of colonoscopy and FIT screening on CRC incidence and stage distribution. The current analysis, which focuses on the impact of colonoscopy and two rounds of FIT screening on CRC incidence and stage, does not appear to substantially advance the existing evidence base.

Although the authors explained their rationale for not conducting a per-protocol analysis, adherence to screening remains a key determinant of screening effectiveness and varies greatly across countries. Low adherence rates inevitably bias the observed effectiveness of screening strategies. This was clearly demonstrated in the NORDICC trial, where variations in participation across countries largely explained the absence of a significant mortality reduction. Importantly, Brenner et al. (Cancer Commun, 2025;45:205–208) reanalyzed the NORDICC data and confirmed that colonoscopy screening significantly reduced CRC incidence and improved early detection, contributing equally to prevention and early diagnosis. These findings highlight that adherence strongly influences the observable effect size of CRC screening.

Reviewer #3

(Remarks to the Author)

In the revised manuscript, the authors have generally addressed my previous concerns. The updated version is improved in clarity. Please find below a few remaining points for consideration:

1. While the authors now state that this is not the interim analysis originally proposed, the Methods/Discussion section would benefit from a more explicit clarification regarding the analytical timeline, specifically: How does this analysis differ from the previously planned but canceled interim analysis?
2. It would be helpful to clearly state how censoring was handled in the Poisson regression models, e.g., whether person-time was truncated at the date of death or migration, and whether any assumptions were assessed or discussed.
3. Adding plots for cumulative incidence/mortality curves would enhance the transparency and interpretability of outcome reporting.

Dear Editor Ming Yang,

We appreciate the opportunity to revise our manuscript titled “Colorectal cancers and adverse events during the diagnostic phase in the SCREEning of Swedish Colons (SCREESCO): A randomized trial comparing primary colonoscopy and fecal immunochemical testing vs usual care”. The editorial comments and the comments from the peer reviewers have helped us improve the manuscript.

Best regards,

Marcus Westerberg, PhD

Comments from the editor

The peer reviewers find the study is of interest. However, they have major concerns with regards to the duration of follow-up, which likely affects the null primary outcome and the conclusions drawn. Previous colorectal cancer screening trials use much longer follow-up duration which is standard to the field. Since the paper reported a cut-off of follow-up on 31 December 2020 (and ~5 years have passed since then), and the trial protocol mentioned plans for reporting longer follow-up, we request that you provide updated results with extended follow-up data.

RE: We thank the editor and the reviewers for the opportunity to clarify the design of SCREESCO, current availability of follow-up and details around the interim analysis mentioned in the protocol (attached as Supplementary Appendix A). We agree with the peer reviewers on most issues and have revised the manuscript accordingly.

Regarding the design: The primary analysis of SCREESCO will be performed at 15 years of follow-up, see trial protocol (original and amended). The final date of follow-up was decided to be on 2030-12-31 (see protocol v3, section 3.8) based on extensive power calculations. The analysis at 15 years includes the primary outcome (CRC-mortality) for which the study was designed (powered), and in addition a description and comparison of CRC incidence.

The interim-analysis: The “plan for longer follow-up” refers to the interim analysis mentioned in the original protocol (v1 section 3.8) and SAP (v1, section “Timing of the analyses”) and concerns CRC mortality at 5/10 years. An additional power calculation (SAP v3, page 6 and Table 3) showed however that there would be limited power to detect differences in CRC-mortality at 10 years (follow-up until 2024-12-31) or earlier. Based on the calculation, the scientific committee decided that this interim analysis should not be performed. With a very limited power, the results would at best be non-informative and at worst misleading and would threaten the integrity of the trial. We have added details on these arguments in the revised manuscript and supplement.

The current analysis: The current analysis of CRCs and adverse events focuses on the diagnostic phase of SCREESCO from 2014 to 2020-12-31 during which the screening and diagnosis was completed. The median follow-up time of 4.8 years allows for medium-term treatment harms to be observed. It also allows for an assessment of the diagnostic yield of both screening and usual care, and specifically whether more CRCs were diagnosed in the

screening arms than in the control arm. We found that incidence rates of stage I-II CRC were indeed higher in the intervention arms than in controls during the first years of follow-up. The incidence rates in the intervention arms became more similar to controls later on during the follow-up.

The current analysis is unique since it provides a comparison of diagnostic-phase outcomes in both the primary colonoscopy arm and the FIT arm (two rounds of two-stool FIT with an unusually low cutoff) with the controls (usual care) using high-quality national registers, overall and separately in men and women. None of the other current trials have a control arm and we are not aware of any previous study comparing overall adverse events in the intervention arm(s) and usual care.

The current assessment of the diagnostic phase of SCREESCO is therefore different to the originally intended interim analysis of the primary outcome (CRC-mortality). We apologize if this has caused confusion and have clarified this in the revised manuscript.

Trial integrity and longer follow-up: The reviewers had concerns about the integrity of the trial. We agree that this is important and hope that our explanations have been clarifying and confirm the integrity of the trial. With the integrity of the trial and aims of the current analysis in mind, prolonging the follow-up would mean that post-intervention data would be published. Long lag times in cancer registration and waiting times for permits to obtain pseudo-anonymized data required for analyses in the current manuscript (including the control group) explain why it took four years from the end of 2020 until we were able to produce the current analysis. We therefore think the current follow-up is most appropriate for reporting the current analysis.

Furthermore, we feel that you need to provide more information regarding the other outcomes, particularly the adenoma incidence– which is requested by one of the reviewers. **RE:** Additional details on the adenomas detected in SCREESCO during the screening phase are included in the revised manuscript. Note that there is no available data in any register on adenomas diagnosed in usual care.

Editorial points to be addressed. Please note that not all of these might apply to your manuscript, in which case please respond with “not applicable.” Structural changes to re-organise the manuscript do not need to be tracked.

* Abstract should be no more than 200 words and for trials should adhere to the CONSORT framework. You must state if the primary outcome was or was not met and provide the effect size and relevant uncertainty estimate. The conclusion of the study must focus only on the primary outcome and safety/tolerability. You must report either all or none of the secondary outcomes, given the space limitation, I would suggest removing all mention of secondary outcomes from the abstract.

RE: We have revised the abstract and reduced the number of words from 300 to 199.

* Please include the trial registration number at the end of the Abstract.

RE: We have included the trial registration number at the end of the revised abstract.

* The Introduction should be written for a broad, non-specialist medical reader and provide sufficient context for the work.

RE: We have revised the introduction accordingly.

* Please provide details of your cohort in the first paragraph of the Results. This includes number of individuals screened for enrolment, as well as exact dates of first and last patient enrolment. The CONSORT patient disposition diagram and the baseline characteristics must be included as main non-text items (for which there is a strict limit of 6 tables and/or figures).

RE: All required details are provided in the first paragraph of the Results.

* The Results should be structured as followed:

- Patient disposition
- Primary outcome(s)
- Secondary outcomes
- Safety
- Exploratory outcomes
- Sensitivity analyses
- Post-hoc analyses

RE: We have revised the Results accordingly. Note that the current analysis is not the final/main analysis of SCREESCO. In the current analysis the outcomes were CRC-diagnoses, adverse events and death from any cause. There were no secondary outcomes, exploratory outcomes or post-hoc analyses.

* Please remove any subheadings from the Discussion.

RE: We have revised the Discussion accordingly.

* Please ensure all results are presented in the Results section, no new data should be introduced in the Discussion.

RE: We have ensured that all results are presented in the Results section.

* You must include explicit paragraphs of study limitations in the Discussion.

* The overarching conclusion of the study must be based only on the primary outcome and safety data.

RE: Limitations are mentioned explicitly in a specific paragraph in the Discussion.

* The Methods should include a full description of the inclusion and exclusion criteria, as well as study procedures and statistical analyses (including a power calculation).

RE: We have revised the Methods section accordingly and included details on the power calculation of SCREESCO with additional details in the amended SAP v3 that is appended.

*Please upload the protocol and SAP with the revision materials, so that reviewers and editors have access to them.

RE: All versions of the protocol and SAP are appended as a supplementary file.

* You must ensure that contributions from all individuals in the author list are available in the Author Contributions statement.

RE: An author Contributions statement can be found at the end of the document.

* Please move all funding sources to the Acknowledgements, including a statement on the

role of the funder.

RE: All info about funding has been moved to the Acknowledgements.

* Please ensure that all potential competing interests are detailed for all authors. For any authors with no competing interests, this must also be stated.

RE: All competing interests have been detailed.

* Please see our guidelines for the Data and Code Availability Statements. “Available on request” is not acceptable, you must provide details of any restrictions to data and code availability. <https://www.nature.com/nature-portfolio/editorial-policies/reporting-standards#availability-of-data>

RE: A “Data sharing statement” is provided at the end of the manuscript file.

* The article file must only contain these items in this order:

- Title
- Author List and affiliations
- Abstract
- Introduction
- Results (with Subheadings)
- Discussion
- Acknowledgements
- Author Contributions
- Competing Interests Statement
- References (for main text only)
- Figure legends (for main text only)
- Tables (note: tables should be pasted into Word files as editable tables, not as images)
- Methods
- Data Availability Statement
- Code Availability Statement
- Methods-only References

RE: The manuscript file has been revised accordingly.

Comments from the reviewers

Reviewer #1 (Remarks to the Author):

Results from the colorectal cancer screening trial SCREESCO diagnostic phase (end of follow-up, December 31, 2020) are presented. While the authors indicate that this is a prespecified analysis, no corresponding details could be found in the protocol. It is unclear whether using a cutoff at 2020, with a median follow-up of 4.8 years, provides a meaningful assessment of incidence endpoints. Longer follow-up may be more informative, especially given that nearly five years have passed since 2020.

RE: We thank the reviewer for the comment. The current analysis of CRCs and adverse events aimed to assess whether more CRCs were diagnosed in the screening arms than in the control arm and whether rates of adverse events were increased or not during the screening phase. The majority of screening colonoscopies were performed within 2 years in the primary colonoscopy arm and within 4 years in the FITx2 arm, and we clearly found an increase of diagnosed stage I-II CRCs compared to usual care during the first years of follow-up. Towards the end of the diagnostic phase, 4-7 years after randomization, we saw that incidence rates of CRC converged. We believe for these reasons that the current follow-up is sufficient for the incidence analysis and hope that this is clear in the revised discussion.

Could the authors comment on potential overdiagnosis, given the elevated number of cancers detected in the colonoscopy arm? Additionally, how are deaths handled when calculating the incidence rates reported in the manuscript?

RE: We thank the reviewer for the comments. We do not expect any overdiagnosis since CRCs were verified histologically in the pathology reports. We have clarified this in the discussion section of the revised manuscript.

Individuals who died during the follow-up window were not considered at risk after date of death in the calculation of incidence rates of CRC and adverse events. We have clarified this in the manuscript and note that the overall mortality rate was similar in all arms.

It may also be valuable to report per-protocol results for comparison. Furthermore, what proportion of participants in the FIT arm actually underwent colonoscopy following a positive FIT test? How are these participants classified in terms of compliance, and if they are not considered noncompliant, could the authors clarify why?

RE: We thank the reviewer for the comments.

The proportion of individuals in the FITx2 arm who performed colonoscopy following positive FIT in each round was around 90%. We have added details on the number/proportion of colonoscopies among positive FITs in each FIT round in a new supplementary figure (**Figure S1**).

Note to editor: Individuals who did not perform a colonoscopy despite a positive FIT, according to the study protocol, are considered non-compliant. We are not sure if this answers the last question above about compliance.

We agree that a per-protocol could be of interest. A naïve per-protocol analysis, however, is biased because participants and nonparticipants are expected to differ in many ways (PMID: 34342631). A more appropriate per-protocol analysis would require more detailed longitudinal data on factors that may affect participation and the outcomes and a more

advanced statistical approach that can integrate these factors longitudinally. We currently do not have longitudinal data in the database on all key factors that would be needed (including health care utilization, granular details on comorbidities and other cancers, etc.) for an informative adjusted per-protocol analysis. A per-protocol analysis is also restricted to those who participated and leads to a smaller sample size and less precision, so such an analysis would be better suited to complement the main analysis at 15 years when more events will have occurred (study end; 2030-12-31).

To that end, we still believe that some descriptive data (baseline characteristics and events during the intervention phase) according to participation/nonparticipation would be of value and have included this information in a new **Supplementary Table S5**. It shows for example that participants differ from non-participants in terms of sex, educational level, origin of birth and comorbidity and we have commented on this in the revised manuscript.

Reviewer #2 (Remarks to the Author):

Thank you for the opportunity to review this important manuscript based on the SCREESCO trial. This is a large-scale randomized controlled study addressing a key question in population-level colorectal cancer (CRC) screening. However, I have several major concerns regarding the current version of the manuscript, which may limit the interpretability, internal validity, and practical implications of the findings.

RE: We thank the reviewer for the comments.

1) The manuscript lacks detailed reporting of participation rates in each screening arm and for each round of FIT screening. Given that screening effectiveness is highly sensitive to uptake, these details are essential for evaluating both internal validity and external generalizability. Some of this information has been previously published (e.g., *Lancet Gastroenterol Hepatol.* 2022;7:513–521), but should be explicitly referenced and integrated into the current manuscript.

RE: We thank the reviewer for this suggestion. We agree and have added additional details on participation in a new **Figure S1** and key info about participation is summarized in the beginning of the Results section. The participation rates are also discussed in the Discussion.

2) Only intention-to-treat (ITT) analyses are presented. Considering the low participation rates reported for both colonoscopy and FIT, ITT analyses alone are insufficient to assess the true effectiveness of the interventions. Per-protocol and as-screened analyses would provide essential complementary information on the potential benefit among participants who actually underwent screening.

RE: We thank the reviewers for the suggestion and agree that it could be of interest. A naïve per-protocol analysis, however, is biased because participants and nonparticipants are expected to differ in many ways (PMID: 34342631). A more appropriate per-protocol analysis would require more detailed longitudinal data on factors that may affect participation and the outcomes and a more advanced statistical approach. We currently do not have longitudinal data in the database on all key factors we think are needed (including health care utilization, fine details on comorbidities and other cancers, etc.) to perform an informative adjusted per-protocol analysis. A per-protocol analysis is also restricted to those who participated and

leads to a smaller sample size and less precision, so it would be better suited for the main analysis at 15 years (study end; 2030-12-31).

To that end, we still believe that some descriptive data (baseline characteristics and events during the intervention phase) according to participation/nonparticipation would be of value and have included this information in a new **Supplementary Table S5**. It shows for example that participants differ from non-participants in terms of sex, educational level, origin of birth and comorbidity and we have commented on this in the revised manuscript.

3) The rationale for using a shared control group across the two screening comparisons (colonoscopy vs usual care and FIT vs usual care) is not clearly explained. This design choice raises questions about comparability and potential statistical dependencies, and should be explicitly justified.

RE: We thank the reviewer for this comment. In the design of the study, power / sample size had to be balanced against ethical aspects, as always. The inclusion of all 60-year-olds in Sweden (except the 3 regions with organized screening) would have yielded the best possible power. The size of the control group had to be minimized, however, to not involve more individuals than necessary but enough such that sufficient power could be obtained. Power would have been somewhat lower at the current sample size if we would have split the control group into two separate groups, one for each of the two intervention arms.

Note that only the FITx2 controls (individuals randomized 2014-2016, during which individuals could be allocated to FITx2) are used in the comparison of FITx2 vs control, while all controls (randomized 2014-2018) are used in the comparison of primary colonoscopy vs control. We do not expect any statistical issues when the two intervention arms are separately compared to their controls, as in the current analysis. Any future direct comparison of the two intervention arms that also utilizes the controls will appropriately account for this design choice.

We have added details on this in the revised supplementary methods of the manuscript.

4) The manuscript focuses on CRC incidence and mortality, but does not report data on advanced adenomas, which are critical intermediate outcomes. Including this information would enhance understanding of the potential for long-term CRC prevention, particularly given the relatively short follow-up time.

RE: We thank the reviewer for the comments and agree. Adenomas were detected and removed at screening colonoscopy in 24% (primary colonoscopy) and 39% (FITx2) of individuals who underwent colonoscopy in SCREESCO. This is on parity with 31% in NordICC. These details on the adenomas detected in SCREESCO are provided in the discussion about potential long-term effects of the prevention. Note that data on adenomas detected in usual care is not available in any national register and only available in the screening colonoscopies of SCREESCO (extensively studied in e.g. PMID 35298893, 38845164 and 36062316).

5) The median follow-up period of 4.8 years is likely insufficient to capture the full preventive effect of screening, particularly for CRC mortality. A longer follow-up (e.g., 10 years) is typically necessary to assess the true impact of screening. The addition of Kaplan–Meier curves for cumulative incidence and mortality would also improve the clarity and transparency of outcome reporting.

RE: We thank the reviewer and agree that a longer follow-up would be able to reveal a potential preventive effect of screening on CRC incidence and mortality, which is the rationale for the trial. In the current analysis, we assessed the diagnostic yield and adverse events during the diagnostic phase in the intervention arms compared to usual care, and in particular whether more CRCs were diagnosed in the screening arms than in the control arm. Such evidence is lacking especially since no other current RCT involving FIT has a control arm. We hope that this is clear in the revised version of the manuscript.

With the integrity of the trial and aims of the current study in mind, we see little added value of prolonging the follow-up and this would delay the publication of the findings of this study by several years due to administrative lag in registers and the lengthy process of obtaining the pseudo-anonymized data. We believe that incidence rates and ratios summarize the findings well.

6) While the manuscript concludes that screening increased detection of early-stage CRC, it does not clearly address the finding that neither screening strategy reduced CRC incidence or mortality within the observed timeframe. Given that incidence was the primary outcome, this null result should be more prominently presented and discussed in the conclusion section.

RE: We thank the reviewer for the comment and agree that a longer follow-up would be able to reveal a potential preventive effect of screening on CRC incidence and mortality, which is the rationale for the trial (with CRC mortality being the main outcome measure of the trial). We have clarified this in the revised discussion section of the manuscript.

7) The limited participation rates and the restriction of the study population to individuals aged 60 years and above should be acknowledged as key limitations. These factors substantially affect the generalizability of the findings and limit their applicability to broader screening populations or policy settings.

RE: We thank the reviewer for the suggestions. The participation rates are mentioned as the first limitation in the paragraph about limitations in the discussion section.

We agree that the restriction to 60-year-olds is a limitation and mention this in the limitations paragraph, but at the same time it can also be seen as a strength, and hence do not regard the age restriction as a key limitation. The reason is that while the restriction to 60-year-olds limits generalizability to other age groups (e.g. 50/70-year-olds) it provides the opportunity of a precise estimate of effect in 60-year-olds. Trials with a wide age range have a more heterogeneous study population with the consequence that the screening effect may vary in subgroups which can be problematic (see e.g. PMID: 39150040).

Reviewer #3 (Remarks to the Author):

In this study, the authors report on the initial diagnostic phase of the SCREESCO trial. This is a large, population-based, randomized controlled trial in Sweden, which was designed to assess the effectiveness of two different colorectal cancer (CRC) screening strategies compared to usual care. In this study, the authors conducted a pre-specified analysis of the trial's diagnostic phase to assess early outcomes. The outcomes for this analysis were the incidence of CRC (overall and by stage), all-cause mortality, and the incidence of gastrointestinal and cardiovascular adverse events. Overall, the study addresses an

important public health question, and the manuscript is generally well-written. However, there are several issues that require attention.

RE: We thank the reviewer for the comments.

1. The most recent Statistical Analysis Plan (version 3.0, 2025-01-16) explicitly states, "The analysis of incidence of colorectal cancer was changed and is now based on cumulative incidence curves instead of the log-rank test." It is not clear whether it is also designed for this prespecified analysis since the details for statistical analyses of this study is not fully described by the protocol. The current manuscript focuses on CRC incidence whereas it does not present cumulative incidence curves. Instead, it reports incidence rates and uses Poisson regression to calculate rate ratios. The authors should address this discrepancy, or they should perform the analysis as specified in the Statistical Analysis Plan (i.e., present cumulative incidence curves and associated statistical tests) and update the manuscript accordingly.

RE: We thank the reviewer for noting this and for allowing us to clarify.

The interim-analysis mentioned in the original protocol (v1 section 3.8) and SAP (v1, section "Timing of the analyses") concerns CRC mortality at 5/10 years. Additional power calculations (SAP v3, page 6 and Table 3) showed however that there would be limited power to detect differences in CRC-mortality at 10 years (follow-up until 2024-12-31) or earlier. Based on the calculations, the scientific committee decided that this interim analysis should not be performed.

The current analysis of CRCs and adverse events focuses on the diagnostic phase of SCREESCO that ended in 2020-12-31. We agree that there could have been more details in the SAP/protocol about the current analysis. To prevent any future confusion regarding the current analysis and the interim analysis mentioned in the SAP/protocol, we have revised the manuscript and supplement, and added details on the reasons for not performing the interim analysis and clarified that the current analysis is not the previously planned interim analysis.

2. The protocol and Statistical Analysis Plan (version 3.0) state, "It was also decided that the previously described interim analysis will not be performed". In Appendix B, they also state "The scientific committee found a need for a timely assessment of baseline findings (diagnosed CRCs and adverse events during the diagnostic phase of the trial) that also includes the control arm (usual care) and diagnoses/events occurring in general (not only those directly related to SCREESCO screening colonoscopies)". Why is the interim analysis canceled, and what is the need for performing this analysis given that the study is powered enough? These should be justified. Also, the authors should clarify in the main manuscript's Methods section that this analysis is not the formal interim analysis that was cancelled. They should clearly state the purpose for this pre-specified analysis to avoid any confusion.

RE: We thank the reviewer for the comment and have clarified in the main manuscript's Methods section that this current analysis is not the formal interim analysis that was not performed because of limited power, and have clarified the purpose of the current analysis. See our answer to the previous comment regarding the reason for not performing the interim analysis. We have also revised the introduction to further clarify the purpose of the current study.

3. While the exclusions of participants post-randomization based on registry linkage are clearly reported (e.g., deaths or CRC diagnoses prior to randomization), it is unclear whether there was any additional loss to follow-up (e.g., due to migration, data linkage failure, or

censoring not at death or event). The authors should explicitly report the number of participants lost to follow-up and provide a brief description of censoring mechanisms.

RE: We thank the reviewer for the comment. We acknowledge that this detail was missing in the manuscript and have revised the methods and results section accordingly. We now indicate the number of individuals who migrated during the follow-up (this is in addition to the two individuals in the control arm that we previously indicated were not identifiable a during data linkage). We have not detected any other reasons for loss of follow-up.

4. Sensitivity analyses should be added. For instance, overdispersion should be assessed for the Poisson models given the large, heterogeneous trial population with regional and clinical variability. If overdispersion is present, the authors may consider addressing it using quasi-Poisson, negative binomial models, or robust standard errors to ensure valid inference.

RE: We thank the reviewer for the suggestion. We estimated dispersion and found that it was within 0.01 of 1.00 for analyses of CRC, and within 0.05-0.1 of 1.00 for analyses of adverse events and death, suggesting no serious deviation from the assumption of mean = variance.

In a sensitivity analysis of adverse events, CIs based on robust standard errors (heteroscedasticity-consistent estimation of the covariance matrix [White's estimator] using the sandwich approach) yielded confidence intervals virtually identical to those of the original analysis.

We have added a comment on this in the methods and results sections.

5. In the main text, it would be helpful to briefly cover what constitutes "usual care" for CRC screening in Sweden for this age group, describing the usual care pathway (e.g., primarily symptomatic detection, availability of opportunistic screening).

RE: We thank the reviewer for the suggestion and agree. We have added the following information on this in the revised manuscript (Methods; new subheading "Usual care" after "Interventions"):

"In Sweden all citizens have access to public health care. A very small minority of individuals have a private health care insurance on top of this (only 0.6% of Swedish healthcare is funded through insurances). During the study period there was no national screening. Screening was performed in the Stockholm region but not in the regions where SCREESCO was performed. In usual care, the main driver of colonoscopies is symptoms. During the study period, FIT has been introduced as an intermediate step in the investigation of symptoms to an increasing extent, where elevated Hb in the fecal sample in a FIT taken because of symptoms triggers a colonoscopy. Individuals who are under surveillance due to increased CRC risk (i.e. former CRC diagnosis, IBD or hereditary/familial CRC syndromes) and may therefore undergo colonoscopy during surveillance. Individuals may also be under surveillance after polypectomy of adenomatous colorectal polyps."

6. I would suggest the authors add explanations for the statistical power of this study because there are several times of adjustments on the sample size due to the lower-than-expectation compliance. For example, in the main text, it was mentioned that the expected statistical power for FITx2 was 80% and 73% for direct colonoscopy. This is somehow weird because the expected effect size for direct colonoscopy should be larger than FIT, though the most recent version of Statistical Analysis Plan has clarified on this.

RE: We thank the reviewer for the comment. The reviewer correctly identified that the assumed effect size was 50% for primary colonoscopy which was higher than 30% for FITx2. The sample size was adjusted once, where two additional years of randomization to primary colonoscopy or control were added. Power is somewhat lower for primary colonoscopy because participation is lower. We have clarified the description of the power calculations in the revised manuscript (subheading “Randomisation, masking and power”).

Dear Editor Ming Yang,

We again appreciate the opportunity to revise and further improve our manuscript titled “Colorectal cancers and adverse events during the diagnostic phase in the SCREening of Swedish Colons (SCREESCO): A randomized trial comparing primary colonoscopy and fecal immunochemical testing vs usual care”.

Best regards,

Marcus Westerberg, PhD

Editorial points

As per the reviewers’ comments, it is very important that you explicitly highlight what is being reported in this trial (that this is an interim analysis at 5 years), and what outcomes are still yet to be reported (that would be reported separately elsewhere, such as the interim analyses of various other time points and primary outcome of colorectal cancer mortality). The reporting of this trial must match what is being reported in the latest version of the clinical trial protocol.

RE: We agree and have clarified which outcomes are still yet to be reported (primary outcome of CRC mortality with follow-up until 2031) and what is reported in this analysis in the revised methods section under the new subheading “**Protocol deviations and rationale for the current study**”. Note that we also list all outcomes analyzed in the current manuscript in the introduction, and that we note in the discussion that CRC mortality will only be reported after the final analysis.

Adverse events is not the primary outcome listed in the protocol and needs to be grouped under “Safety”.

RE: We have revised the manuscript accordingly.

In the discussion- please remove descriptive statistics from other studies- and the discussion needs to be framed around big-picture points on the potential impact of this screening trial in informing colorectal cancer screening guidelines. You need to also convey the conceptual advance of this trial (i.e. why is including a no-screening control arm important)

RE: We have removed descriptive statistics from the paragraph about other studies and have emphasized the big picture by removing minor comments on sex differences. We have also highlighted the big-picture point regarding CRC (new text in **bold**):

“In the current study the rates of stage I-II cancer were higher in both intervention arms compared to controls, in particular during the first years of follow-up when most of the screening colonoscopies were performed. **This excess risk decreased later on, and the incidence of stage III-IV simultaneously decreased after around 4 years in particular in the FIT arm compared to controls. It is likely too early to detect a net benefit of the prevention in terms of a lower CRC incidence in the intervention arms compared to controls and/or determine if (a part of) the early-stage CRC excess risk in the intervention arms represents an overdiagnosis of clinically insignificant CRCs.**”

We have also added the following to the discussion (new text in **bold**) to highlight the conceptual advance:

“The only ongoing randomized controlled trials other than SCREESCO, involving both primary colonoscopy and FIT, are the Spanish trial COLONPREV and the American trial Colonoscopy vs. Fecal Immunochemical Test in Reducing Mortality From Colorectal Cancer (CONFIRM), none of which has a control arm. **SCREESCO is unique in providing a comparison of screening effectiveness between invitation to FIT screening and usual care.**”

Please address the following editorial and formatting points:

- We do not allow “code available upon request”. Please either deposit the code in a repository, such as GitHub and provide the appropriate details. Alternatively, please provide more information for readers on any restrictions to accessing the code and how the code can be accessed.

RE: The analytical code has been uploaded to GitHub and a link has been provided in the revised version of the manuscript.

- Any references cited only in the methods needs to be included in a separate methods-only references section and should be numbered contiguously to the main reference list (i.e. number starts at XX following on from the numbering of the main reference list, not 1).

RE: The manuscript has been revised accordingly.

- Please be aware that we are unable to accommodate supplementary text. Please integrate them into either the main text or the methods section. Please note that there is no limit for our Methods section as it is online only.

RE: The supplementary text has been integrated into the methods section.

- The supplementary material cannot include references. Please move the references in the supplementary material to the main reference list and ensure that this is appropriately cited in the main text.

RE: The supplementary references have been integrated into the methods-only references.

- We do not allow supplementary Figures. Please convert supplementary figure 1 (Figure S1) into an extended display item (Extended data figure 1)

RE: The supplementary figure has been uploaded as **Extended data figure 1**.

- Please organise your authorship list so that the list of authors and affiliations are listed separately, with the affiliations as a numbered list (e.g. Marcus Westerberg (1), Jonas F. Ludvigsson (2,3))

RE: We have revised the manuscript accordingly.

- Please do the same for your SCREESCO principle investigators as with your primary authorship list- I believe you want these PIs to be part of the authorship list as a consortium?

RE: We do not want these to be considered a part of the authorship list and have removed the list of principal investigators from the supplement.

- Please remove other details like Word count, word count abstract, tables and figures, references numbers, keywords etc (everything in Page 3)

RE: We have revised the manuscript accordingly.

- The funding information in page 4 needs to go to the Acknowledgements.

RE: We have revised the manuscript accordingly.

- Ethics approval needs to be moved to the Methods

RE: We have revised the manuscript accordingly – ethical approvals are mentioned in the second paragraph of the methods section.

- You need a separate section in the Methods detailing all protocol deviations

RE: We have added a subsection “**Protocol deviations and rationale for the current study**” to the Methods detailing all protocol deviations and the rationale for the current study based on our previous responses and text from the previous supplementary methods.

Other editorial points to be addressed. Please note that not all of these might apply to your manuscript, in which case please respond with “not applicable.” Structural changes to re-organise the manuscript do not need to be tracked.

* Abstract should be no more than 200 words and for trials should adhere to the CONSORT framework. You must state if the primary outcome was or was not met and provide the effect size and relevant uncertainty estimate. The conclusion of the study must focus only on the primary outcome and safety/tolerability. You must report either all or none of the secondary outcomes, given the space limitation, I would suggest removing all mention of secondary outcomes from the abstract.

RE: Done, 199 words.

* Please include the trial registration number at the end of the Abstract.

RE: Done!

* The Introduction should be written for a broad, non-specialist medical reader and provide sufficient context for the work.

RE: Done!

* Please provide details of your cohort in the first paragraph of the Results. This includes number of individuals screened for enrolment, as well as exact dates of first and last patient enrolment. The CONSORT patient disposition diagram and the baseline characteristics must be included as main non-text items (for which there is a strict limit of 6 tables and/or figures).

RE: Done! Consort diagram in Figure 1 and baseline characteristics in Table 1.

* The Results should be structured as followed:

- Patient disposition
- Primary outcome(s)
- Secondary outcomes

- Safety
- Exploratory outcomes
- Sensitivity analyses
- Post-hoc analyses

RE: Done!

* Please remove any subheadings from the Discussion.

RE: Done!

* Please ensure all results are presented in the Results section, no new data should be introduced in the Discussion.

RE: Done!

* You must include explicit paragraphs of study limitations in the Discussion.

RE: Done!

* The overarching conclusion of the study must be based only on the primary outcome and safety data.

RE: Done!

* The Methods should include a full description of the inclusion and exclusion criteria, as well as study procedures and statistical analyses (including a power calculation).

RE: Done!

* Please upload the protocol and SAP with the revision materials, so that reviewers and editors have access to them.

RE: Included in the submission as supplementary information.

* You must ensure that contributions from all individuals in the author list are available in the Author Contributions statement.

RE: Done!

* Please move all funding sources to the Acknowledgements, including a statement on the role of the funder.

RE: Done!

* Please ensure that all potential competing interests are detailed for all authors. For any authors with no competing interests, this must also be stated.

RE: Done!

* Please see our guidelines for the Data and Code Availability Statements. "Available on request" is not acceptable, you must provide details of any restrictions to data and code availability. <https://www.nature.com/nature-portfolio/editorial-policies/reporting->

standards#availability-of-data

RE: R code is available on GitHub and a link has been included in the revised manuscript.

* The article file must only contain these items in this order:

- Title
- Author List and affiliations
- Abstract
- Introduction
- Results (with Subheadings)
- Discussion
- Acknowledgements
- Author Contributions
- Competing Interests Statement
- References (for main text only)
- Figure legends (for main text only)
- Tables (note: tables should be pasted into Word files as editable tables, not as images)
- Methods
- Data Availability Statement
- Code Availability Statement
- Methods-only References

RE: We have revised the manuscript accordingly.

Reviewers' Comments

Reviewer #1 (Remarks to the Author):

Histologic confirmation rules out false positives but does not eliminate the possibility of overdiagnosis, as some pathologically confirmed cancers may still be slow-growing and clinically insignificant. The increased number of early-stage cancer diagnoses in the intervention arm could therefore reflect overdiagnosis. Particularly if this rise is not accompanied by a corresponding decline in late-stage or metastatic cancers and cancer-related deaths, which is the hallmark of overdiagnosis. A reduction in late-stage cancer incidence may serve as a more meaningful intermediate outcome. However, it is unclear whether the current diagnostic-phase results, based on a median follow-up of 4.8 years, are sufficient to yield meaningful conclusions.

RE: We thank the reviewer for the comment. We agree in principle that some pathologically confirmed CRCs may be slow-growing and clinically insignificant, although the risk of overdiagnosis is lower for CRCs than overtreatment of precancerous polyps (PMID: 30076834). We expect that this risk of overdiagnosis of clinically insignificant CRCs may apply especially to older individuals that have shorter life expectancy than younger individuals aged around 60 years. The final analysis of the trial in 2031 will be able to answer whether there is a reduction in late-stage CRC incidence and CRC mortality.

We have clarified our interpretation of the results in the revised discussion section (new text in **bold**):

“It is likely too early to detect a net benefit of the prevention in terms of a lower CRC incidence in the intervention arms compared to controls **and/or determine if (a part of) this increase of early-stage CRCs in the intervention arms corresponds to an overdiagnosis of clinically insignificant CRCs.**”

Reviewer #2 (Remarks to the Author):

The SCREESCO study indeed represents an important randomized controlled trial (RCT) in the field of colorectal cancer (CRC) screening, and the authors have made efforts to address several points raised by the reviewers. However, despite these revisions, the current manuscript provides only limited additional evidence beyond what is already available from recent large-scale RCTs.

Two major RCTs on CRC screening have been published recently — the COLOPREV trial (Lancet. 2025;405:1231–1239) and the NORDICC trial (N Engl J Med. 2022;387:1547–1556). Both provide robust data on the effects of colonoscopy and FIT screening on CRC incidence and stage distribution. The current analysis, which focuses on the impact of colonoscopy and two rounds of FIT screening on CRC incidence and stage, does not appear to substantially advance the existing evidence base.

Although the authors explained their rationale for not conducting a per-protocol analysis, adherence to screening remains a key determinant of screening effectiveness and varies greatly across countries. Low adherence rates inevitably bias the observed effectiveness of screening strategies. This was clearly demonstrated in the NORDICC trial, where variations in participation across countries largely explained the absence of a significant mortality reduction. Importantly, Brenner et al. (Cancer Commun, 2025;45:205–208) reanalyzed the NORDICC data and confirmed that colonoscopy screening significantly reduced CRC incidence and improved early detection, contributing equally to prevention and early diagnosis. These findings highlight that adherence strongly influences the observable effect size of CRC screening.

RE: We thank the reviewer for the comment and agree that adherence strongly influences the observable effect size of CRC screening. We plan to update the study protocol and SAP with a detailed pre-specification of per-protocol analyses to be performed with follow-up until 2031-12-31. We have also added some details to the limitations of the current study regarding the implications of adherence in SCREESCO in the revised limitations paragraph in the discussion section:

“... **We therefore likely observed a lower increase in CRC incidence during the diagnostic phase compared to what would be expected in settings with higher adherence.**”

Reviewer #3 (Remarks to the Author):

In the revised manuscript, the authors have generally addressed my previous concerns. The updated version is improved in clarity. Please find below a few remaining points for consideration:

1. While the authors now state that this is not the interim analysis originally proposed, the Methods/Discussion section would benefit from a more explicit clarification regarding the

analytical timeline, specifically: How does this analysis differ from the previously planned but canceled interim analysis?

RE: We thank the reviewer for the comment and have clarified how the current analysis is different to the originally proposed interim analysis in the revised methods section that now includes information from the now removed supplementary methods. This new subsection is called “Protocol deviations and rationale for the current study” and reads:

The primary outcome in SCREESCO is mortality from CRC at 15 years. Power calculations were performed for this outcome alone. The initial protocol and current protocol (Supplementary Note 1) state, however, that the main research questions of SCREESCO are to investigate 1) if screening has an effect on the mortality from CRC, 2) if screening has an effect on the incidence of CRC, and 3) what method should be used in Sweden regarding the effect according to (1) and (2). The scientific committee found a need for a timely assessment of baseline findings (diagnosed CRCs and adverse events during the diagnostic phase of the trial) that also includes the control arm (usual care) and diagnoses/events occurring in general (not only those directly related to SCREESCO screening colonoscopies). The current study is therefore listed as a planned main study (Section 9, Study protocol version 3 in Supplementary Note 1). Note that CRC mortality is not assessed in the current study and that this main outcome of the trial will instead be presented in the final report of the trial with follow-up until 2031-12-31.

Note that this study does not constitute the interim analyses mentioned of CRC mortality at 5 and 10 years of follow-up in the original study protocol and SAP. An additional power calculation (SAP v3, page 6 and Table 3 in Supplementary note 1) showed that there would be limited power to detect differences in CRC-mortality at 10 years (follow-up until 2024-12-31) or earlier. Based on the calculation, the scientific committee decided that that interim analyses of CRC mortality should not be performed.

The current study focuses on early CRCs and adverse events during the diagnostic phase is listed as a planned main study publication. It builds on a previous baseline assessment of screen-detected CRCs and adverse events in screening colonoscopies of SCREESCO1 and can be considered an extended baseline report or a 5-year interim analysis of diagnosed CRCs and adverse events occurring during the intervention phase. It includes extended information from Swedish health registers regarding CRCs diagnosed and adverse events in general in both screening arms and in the control arm during the diagnostic phase of SCREESCO when screening FITs and colonoscopies were performed.

2. It would be helpful to clearly state how censoring was handled in the Poisson regression models, e.g., whether person-time was truncated at the date of death or migration, and whether any assumptions were assessed or discussed.

RE: We thank the reviewer for the comment and have clarified how censoring was handled in the revised methods section:

“Individuals were censored when they migrated out of Sweden before the outcome had been observed (if this occurred). Individuals were similarly not considered at risk after their date of death. We assessed potential violations of equidispersion for each regression model and performed a sensitivity analysis where robust standard errors were used to compute the confidence intervals.”

3. Adding plots for cumulative incidence/mortality curves would enhance the transparency and interpretability of outcome reporting.

RE: We thank the reviewer for the suggestion and have after checking with the editor decided to include an extended data figure that shows cumulative incidence of CRC in all arms (overall and by risk stage).

We have updated the results section accordingly and added the following text to the methods section:

“In a complementary analysis we computed competing risk cumulative incidence curves of CRC (overall and by stage) where death from any cause was considered a competing risk.”